# Stromal-driven and Amyloid β-dependent induction of neutrophil extracellular traps modulates tumor growth

Hafsa Munir[1], James O. Jones[1,2], Tobias Janowitz[3,4,5], Markus Hoffmann [6], Maximilien Euler[6], Carla P. Martins[1,7], Sarah J. Welsh[2,5] & Jacqueline D. Shields [1✉]

Tumors consist of cancer cells and a network of non-cancerous stroma. Cancer-associated fibroblasts (CAF) are known to support tumorigenesis, and are emerging as immune modulators. Neutrophils release histone-bound nuclear DNA and cytotoxic granules as extracellular traps (NET). Here we show that CAFs induce NET formation within the tumor and systemically in the blood and bone marrow. These tumor-induced NETs (t-NETs) are driven by a ROS-mediated pathway dependent on CAF-derived Amyloid β, a peptide implicated in both neurodegenerative and inflammatory disorders. Inhibition of NETosis in murine tumors skews neutrophils to an anti-tumor phenotype, preventing tumor growth; reciprocally, t-NETs enhance CAF activation. Mirroring observations in mice, CAFs are detected juxtaposed to NETs in human melanoma and pancreatic adenocarcinoma, and show elevated amyloid and β-Secretase expression which correlates with poor prognosis. In summary, we report that CAFs drive NETosis to support cancer progression, identifying Amyloid β as the protagonist and potential therapeutic target.

[1] MRC Cancer Unit, Hutchison/MRC Research Centre, University of Cambridge, Box 197 Cambridge Biomedical Campus, Cambridge CB2 0XZ, England. [2] Department of Oncology, Cambridge University Hospitals NHS Foundation Trust, Cambridge Biomedical Campus, Hills Road, Cambridge CB2 0QQ, England. [3] Cold Spring Harbor Laboratory, Cold Spring Harbor, New York, NY 11724, USA. [4] Northwell Health Cancer Institute, New York, NY 11021, USA. [5] Cancer Research UK Cambridge Institute, University of Cambridge, Li Ka Shing Centre, Cambridge CB2 0RE, UK. [6] Friedrich Alexander University Erlangen-Nuremberg, Universitätsklinikum Erlangen, Department of Medicine 3, Universitätsstrasse 25a, 91054 Erlangen, Germany. [7] Present address: Early Oncology TDE, Oncology R&D, AstraZeneca, Cambridge CB2 0RE, England. ✉email: js970@mrc-cu.cam.ac.uk

The tumor microenvironment comprises a complex niche of cancer cells and "normal" cell populations collectively referred to as the stroma. The stroma includes leukocytes, cancer-associated fibroblasts (CAFs), pericytes, blood vessels and lymphatic vessels[1]. Tumor development is accompanied by changes in the phenotype, function and interactions between these stromal constituents[2–5], a process which is central to carcinogenesis.

CAFs are one of the most abundant stromal populations in the tumor, and display considerable heterogeneity and plasticity[6]. Multiple tumor-promoting functions have been attributed to CAFs, including promoting angiogenesis, remodeling extracellular matrix[7,8], modifying tumor stiffness[9,10], nutrient processing[11], and facilitating the invasion of tumor cells[1,4]. More recently, CAFs have emerged as modulators of the innate and adaptive immune responses as they are recruited to the tumor. CAF-derived factors have been shown to drive the recruitment and M2 polarization of macrophages[12]. CAFs drive T-cell deletion and exhaustion in an antigen-dependent manner by FASL and PD-L2 mediated interactions[5]. CAFs also mediate exclusion of T-cells from tumors via the production of CXCL12[2]. Furthermore, CAF-derived IL-6 can induce systemic immunosuppressive effects[13]. Collectively, these data demonstrate that anti-cancer immune responses can be modulated by CAFs, and failure of the immune system to control cancer is not solely mediated by cancer cells but also the surrounding stroma.

Neutrophils are the most abundant circulating leukocyte population, functioning as early responders to inflammatory insult[14]. Following activation, neutrophils utilize several mechanisms to exert their effects, such as secreting inflammatory factors that influence other immune populations, production of reactive oxygen species (ROS) and cytotoxic granular proteins to eliminate pathogens, as well as releasing extracellular traps (NETs). NETs are composed of chromatin-bound DNA, decorated in cytosolic and granular proteins such as myeloperoxidase (MPO) and neutrophil elastase (NE)[14]. Though the molecular mechanisms governing NET release are still not completely understood, in certain contexts it requires ROS-mediated, calcium driven citrullination of histones by Protein Arginine Deiminase 4 (PAD4)[14].

NETs detected in chronic inflammatory disorders exert their pro-inflammatory effects by modulating the activity of other stromal populations at the site of tissue damage. In murine models of inflammation, such as systemic lupus erythematosus, NETs activate plasmacytoid dendritic cells through engagement of TLR9, thus exacerbating the condition[15,16]. In addition, NETs have been shown to reduce the threshold for the activation of CD4[+] T-cells in response to antigen[17]. As well as regulating immune cells, NET-derived components were found to induce activation of lung fibroblasts in a lung fibrosis model, promoting their differentiation into myofibroblasts[18]. Subsequent collagen deposition, proliferation and migration of the differentiated myofibroblasts was also enhanced by treatment with NETs[18].

While the roles of neutrophils in infection have been well established, their contribution to tumor progression, immune evasion and metastasis remain controversial. Indeed, neutrophils have been reported to exert both anti- and pro-tumorigenic effects depending on the environmental cues to which they are exposed[19,20]. This plasticity has made it difficult to ascertain the spatial and temporal nature of neutrophil functions within tumors. NETs have been identified as modulators of cancer-induced thrombosis through a granulocyte-colony-stimulating factor (G-CSF) dependent mechanism[21] and facilitate metastasis by capturing circulating tumor cells to promote colonization of distal sites[22–25]. They have also been detected in several blood cancers[26–28] and recently, NET-mediated remodeling of the extracellular matrix has been reported to awaken dormant cancer cells and promote aggressive lung metastasis[29]. While observed in primary tumors, the contribution of NETs to tumorigenesis and underlying mechanisms are lacking[22,30–34].

Here, we explore the interactions between CAFs and tumor-infiltrating neutrophils in three different models of cancer, determining the effects of CAF-derived factors on the capacity of neutrophils to undergo NETosis and how this process influences tumor growth. We show that CAF-secreted Amyloid β drives formation of tumor-associated NETs (t-NETs) through CD11b in a ROS-dependent mechanism both within the microenvironment and at systemic levels in the blood and bone marrow. Therapeutic inhibition of t-NETosis, or prevention of Amyloid β release by CAFs abolishes growth of established tumors and restores an anti-tumor status in the tumor-infiltrating neutrophils, indicating a potential axis to be exploited therapeutically in multiple cancer types.

## Results

While neutrophils are observed in tumors, little is known about how micro-niches created by stromal compartments perturb the activity of these immune cells. In primary murine pancreatic and skin (melanoma) tumors we observed that neutrophils were frequently confined to CAF-rich regions (Fig. 1A) implying a potential for cross-talk between the two populations.

**Influence of CAF-derived factors on neutrophil function**. To determine if CAF-derived factors have the capacity to impact aspects of neutrophil behavior, we first treated bone marrow-derived neutrophils with conditioned media (CMed) from CAFs, tissue-matched normal FBs, or Phorbol 12-myristate 13-acetate (PMA; a well-known activator of neutrophils) and assessed viability, surface activation markers, ROS production and phagocytic capability of the neutrophils. Isolated CAFs were characterised based on surface expression of classic markers; Thy1, Podoplanin and PDGFRα (Supplementary Fig. 1A) and lack of immune (CD45) and epithelial (EpCAM) markers (Supplementary Fig. 1A). Lung and pancreatic FB or CAF CMed had no impact on neutrophil viability for the duration of treatment while PMA induced low levels of neutrophil death (Supplementary Fig. 1B). Expression of activation markers (CD11b and CD18) tended to increase in response to lung CAF CMed treatment (Supplementary Fig. 1C) in the presence or absence of an additional inflammatory insult (Lipopolysaccharide; LPS). CAF CMed failed to induce ROS production in neutrophils (Supplementary Fig. 1D) nor did the cells exhibit enhanced phagocytic capabilities (Supplementary Fig. 1E) relative to PMA after 30 min treatment. Therefore, the classical functions of neutrophils[14] were largely unaltered by treatment with CAF-derived factors in vitro.

**CAF-derived factors induce NETs in primary tumors**. Having already observed neutrophils in proximity to CAF-rich regions, we also detected the presence of NETs within primary murine pancreatic, skin and lung tumors (Fig. 1B and Supplementary Fig. 1F). NETs were defined as staining positive for extracellular DNA, MPO and Citrullinated histone H3 (CitH3). We termed these structures tumor-induced NETs (t-NETs), and thus sought to examine the role of t-NETs in the primary tumor, focusing on the mechanisms driving their release.

To determine if CAF-derived factors drive the generation of t-NETs, we treated isolated bone marrow neutrophils with CAF or FB CMed and analyzed NETosis (Supplementary Fig. 2A, B and Supplementary Movie 1). While CMed from normal FBs was unable to induce NETs, CAF CMed from pancreatic, lung and skin tumors was sufficient to induce NETosis to levels comparable

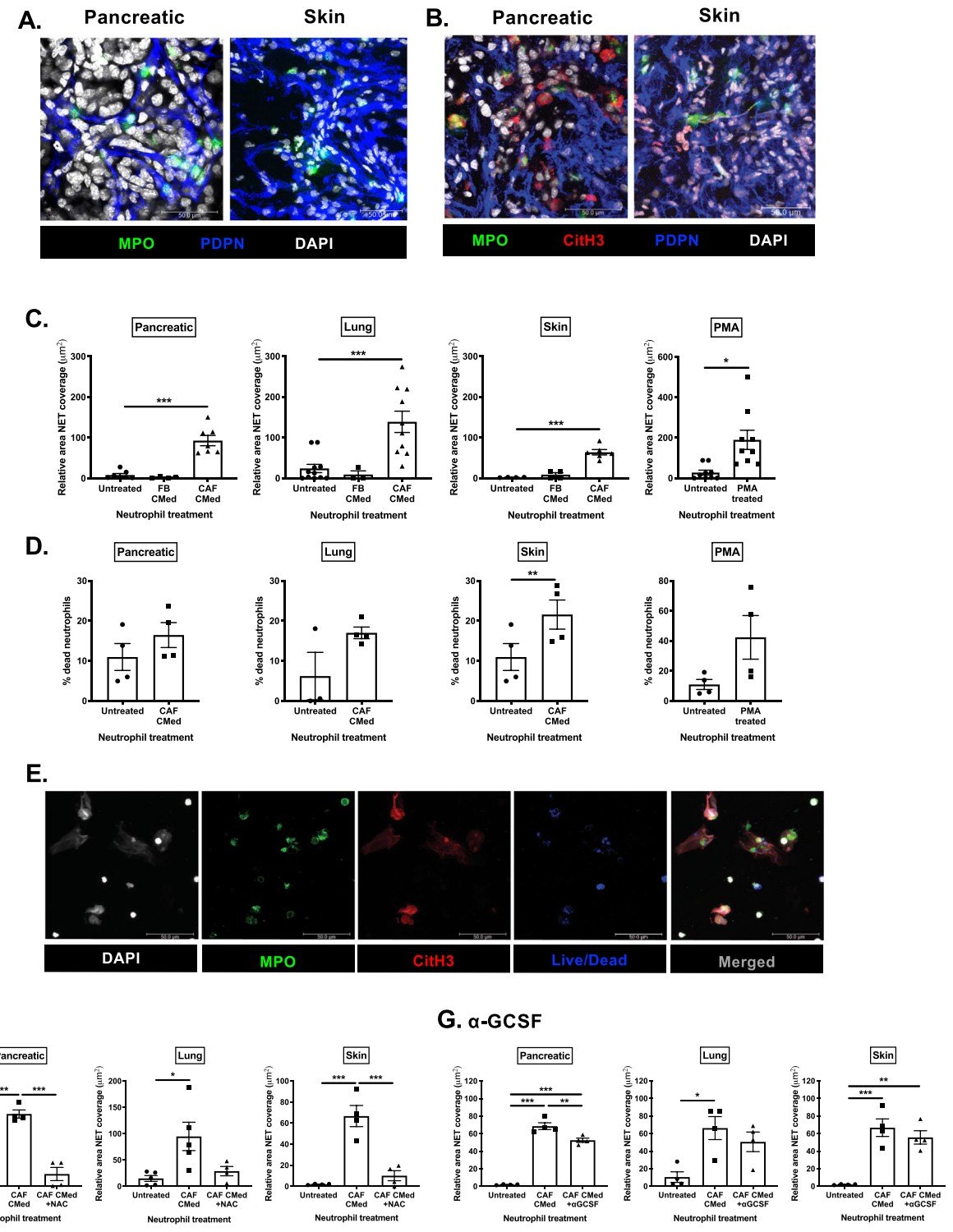

with PMA (Fig. 1C and Supplementary Fig. 2B). As NETosis is thought to be largely a "suicidal" process[14], we next quantified neutrophil death following longer-term exposure to CAF CMed (compared to 30 min treatment in Supplementary Fig. 1B). After 3 h of treatment with CAF CMed, a trend of increasing cell death was observed compared to untreated neutrophils and to levels comparable with PMA (Fig. 1D). This was confirmed by staining of NETting neutrophils with live/dead dye (Fig. 1E) indicating that CAF-influenced NETosis is a cell-death dependent effect. As autophagy has also been implicated as a mode of death in

NETosis[34], we then quantified the level of NETosis in the presence of chloroquine, an inhibitor of autophagy. Chloroquine did not rescue the neutrophils from undergoing NETosis (Supplementary Fig. 2C) ruling out autophagy as a mechanism underlying CAF-induced t-NET formation.

G-CSF and intracellular ROS have been reported to induce citrullination of Histone H3 which is required for NET formation[35]. Thus, to determine if CAF-derived factors stimulate t-NETs in a ROS-dependent manner, we first inhibited ROS production. Pre-treatment of neutrophils with N-acetylcysteine

**Fig. 1 CAF-derived factors induce NETosis in vitro and in vivo. A** Confocal microscopy of myeloperoxidase (MPO) and podoplanin (PDPN) expressed on neutrophils and CAFs respectively in pancreatic, skin tumors. **B** Confocal microscopy of NETs in murine pancreatic, skin tumors showing expression of myeloperoxidase (MPO) and Citrullinated histone H3 (CitH3) by NETting neutrophils and podoplanin (PDPN) by CAFs. **C** Quantification of the area of the field covered by SYTOX green positive neutrophil-derived extracellular DNA relative to the number of neutrophils in each field after treatment with pancreatic, lung or skin FBs CMed, CAF CMed or PMA for 3 h. **D** The percentage of dead neutrophils after treatment with pancreatic, lung or skin CAF CMed or PMA based on the number of SYTOX green positive neutrophils. **E** Confocal microscopy of bone marrow neutrophils stained with MPO, CitH3 and live/dead cell viability dye after induction of NETosis by treatment with CAF CMed for 3 h. Quantification of the relative NET coverage of neutrophils treated with CAF CMed with or without pre-treatment with **F** N-acetyl-cysteine (NAC) or **G** anti-granulocyte colony-stimulating factor (α-GCSF). Data are mean ± SEM; *$p < 0.05$, **$p < 0.01$ and ***$p < 0.001$ using (**C**, **F** and **G**) one-way ANOVA with a Tukey post hoc test (with the exception of 1C, untreated vs PMA, which was performed with a paired $t$-test) and (**D**) paired $t$-test. Assays were performed on (**A–B**) Representative images of $n = 6$ tumors, (**C**) $n = 7$, 4 and 7 (Pancreatic for unstimulated, FB CMed and CAF CMed treated, respectively), $n = 11$, 3 and 10 (Lung for unstimulated, FB CMed and CAF CMed treated, respectively), $n = 4$, 4 and 6 (Skin for unstimulated, FB CMed and CAF CMed treated, respectively) and $n = 9$ (PMA), (**D**) $n = 4$, (**E**) $n = 2$, **F** $n = 4$ (Pancreatic and Skin) $n = 5$ (Lung) and (**G**) $n = 4$ independent experiments. Scale bars are 50 μm.

(NAC; Fig. 1F) Diphenyleneiodonium (DPI), Trolox or Vitamin C (Supplementary Fig. 2D–F) prior to CAF CMed treatment suppressed NET formation, indicating that ROS production by neutrophils in response to CAF-derived factors is a key mediator in the process. However, unlike previous reports which showed that tumor-derived G-CSF drives NETosis systemically, here CAF-mediated NETosis was not driven by G-CSF as NET release was not significantly reduced following neutralization of G-CSF in vitro (Fig. 1G). Together, these data suggest that CAFs are key drivers of ROS-dependent, suicidal t-NETosis and the potential factors that induce t-NETs in this context are likely distinct from those previously reported[21,34].

**CAF-derived factors induce systemic effects on neutrophils.** We next investigated whether CAF-derived factors could render circulating neutrophils more susceptible to NETosis before being recruited into the tumor. Indeed, we observed that bone marrow-derived neutrophils isolated from pancreatic, lung and skin tumor-bearing mice displayed a greater propensity to generate t-NETs in the absence of an additional stimulus compared to neutrophils from non-tumor-bearing mice (Fig. 2A). With pancreatic tumors, but not for either lung or skin, this was accompanied by an increase in neutrophil death in the bone marrow (Supplementary Fig. 3A). However, there was not a significant increase in the number of neutrophils isolated from tumor-bearing mice (Fig. 2B).

To determine whether CAF-derived factors were sufficient to drive the observed susceptibility of bone marrow-derived neutrophils towards NETosis, we intravenously (I/V) infused CMed from pancreatic or lung-derived FBs or CAFs in the absence of tumors. Spontaneous NET production by bone marrow-derived neutrophils ex vivo was significantly enhanced in wild-type mice treated with CAF CMed compared to FB CMed or basal media (Fig. 2C) with concurrent increases in neutrophil death (Supplementary Fig. 3B) and counts (Fig. 2D). The increases in neutrophil number observed following CAF and FB CMed infusion indicates that FB and CAFs may also be a source of additional factors, such as G-CSF capable of boosting neutrophil production. Discrepancies between neutrophil counts after I/V infusion of the CMed vs. tumor-bearing mice may reflect a difference in the concentration of factors capable of driving neutrophil expansion in CMed infused compared to the amount secreted by the tumor stroma.

**CAF-driven t-NETs are pro-tumorigenic.** Having shown that CAFs can promote t-NETosis at local and systemic levels we next sought to determine the functional impact of the t-NETs on primary tumor development. Previous studies have analyzed the effect of neutrophil depletion on tumor progression, primarily by

using anti-Ly6G antibodies[36–38], however when studying NETosis this approach would have confounding effects as a result of their depletion. Therefore, to disrupt NETosis without impacting other neutrophil functions, we inhibited PAD4 which drives citrullination of histones to facilitate DNA release[14]. To first ascertain the requirement of PAD4 in CAF-driven NETosis, we tested the effects of its inhibition on neutrophils in vitro. A complete suppression of CAF CMed-induced NET release was observed in neutrophils treated with the PAD4 inhibitor, Cl-amidine, in vitro (Fig. 2E). Moreover, treatment with Cl-amidine in vivo prior to I/V infusion of lung CAF CMed was sufficient to abolish NET release by bone marrow-derived neutrophils along with a reduction in neutrophil death ex vivo (Fig. 2F and Supplementary Fig. 3C). The number of bone marrow neutrophils was largely unaffected (Fig. 2G) suggesting that NETosis was inhibited without influencing other factors driving an increase in neutrophil number.

In vivo, we focused on the effect of blocking t-NETs on the development of primary skin and pancreatic tumors. Lung adenocarcinomas were not examined following observations that they only developed a significant CAF compartment when the tumors were higher grade, thus confounding our ability to determine the importance of CAF-driven NETosis on tumor progression. Mice bearing established melanoma were treated with GSK484, Cl-amidine (PAD4 inhibitors) or DMSO for 7 days (Fig. 3A and Supplementary Fig 4A). Cl-amidine treatment completely inhibited tumor growth compared with vehicle controls (Supplementary Fig 4A). This stasis effect was replicated following treatment with the more specific PAD4 inhibitor, GSK484 (Fig. 3B). While tumor volumes were drastically different following inhibition of t-NETosis, we did not observe a significant effect on tumor immune infiltrates and resident stroma by flow cytometry profiling (Fig. 3C and Supplementary Fig. 4B, C, E). Importantly, the number of neutrophils recruited to tumors was unaffected by PAD4 inhibition, indicating that treatment effects on neutrophil behavior were specific to NETosis at the tumor (Fig. 3C, D and Supplementary Fig. 4B). Suppression of tumor growth was not due to direct toxicity of the small molecule inhibitors on cancer cells. Indeed, treatment with GSK484, had no effect on the growth of melanoma cells in vitro (Supplementary Fig. 4F), while treatment with Cl-amidine only mildly affected cell growth (non-significant; Supplementary Fig. 4F) which could be a potential consequence of the broader action of Cl-amidine.

With the knowledge that circulating NETs have been reported to contribute to thrombus formation in advanced disease, we also examined the plasma of mice treated with PAD4 inhibitors. GSK484, but not Cl-amidine treatment was accompanied by lower levels of the clotting factors von willebrand factor (vWF),

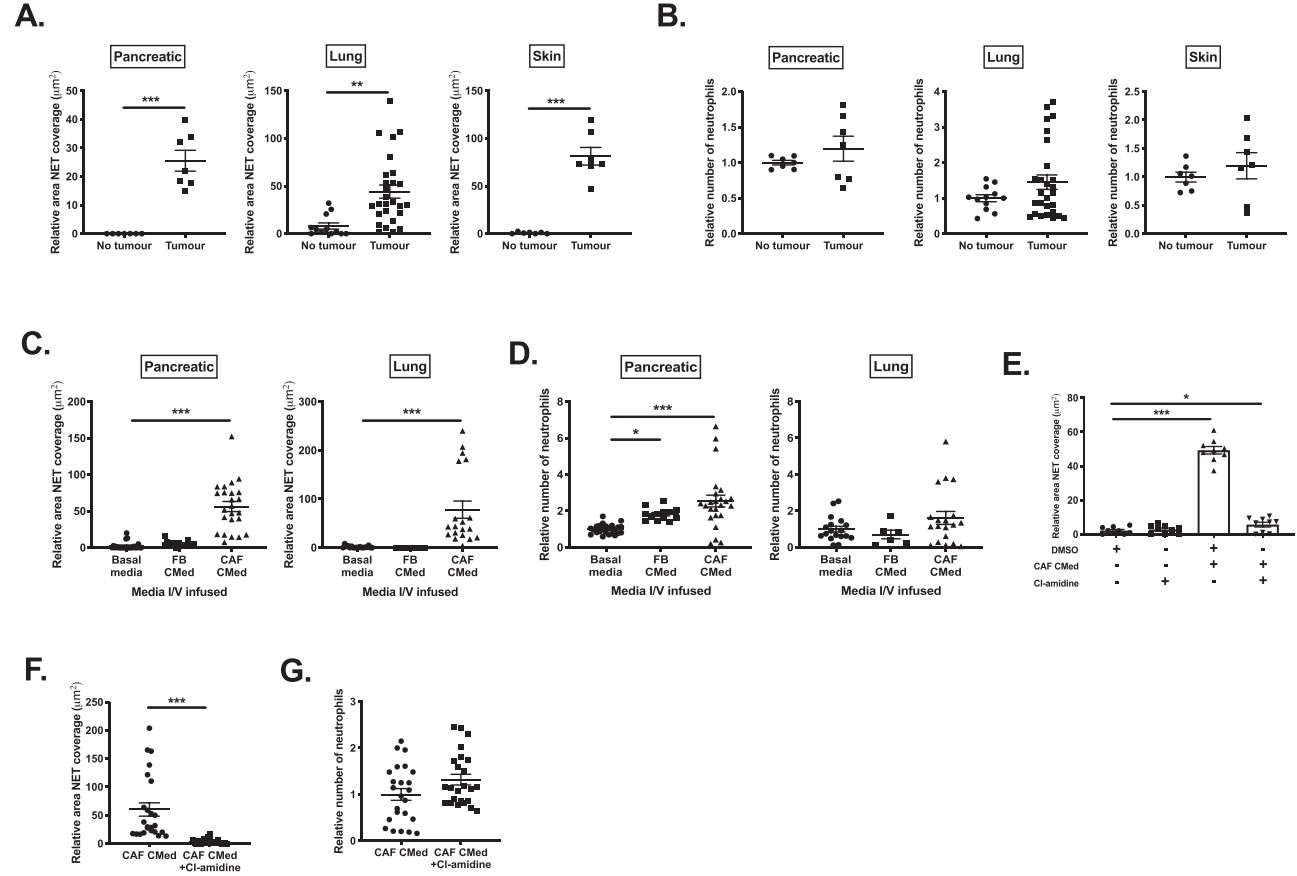

**Fig. 2 CAF-derived factors drive NETosis systemically.** Quantification of **A** relative NET coverage and **B** relative number of bone marrow neutrophils taken from mice with pancreatic, lung or skin tumors. Quantification of the **C** relative NET coverage and **D** relative number of bone marrow neutrophils isolated from wild-type mice intravenously infused with pancreatic or lung FB or CAF CMed 24 h before analysing NETosis. **E** Quantification of relative NET coverage after stimulation with pancreatic CAF CMed with or without treatment with Cl-amidine in vitro. **F** Quantification of the relative NET coverage and **G** relative number of bone marrow neutrophils isolated from mice intravenously infused with lung CAF CMed with or without pre-treatment with Cl-amidine for 24 h. Data are mean ± SEM; *$p < 0.05$, **$p < 0.01$ and ***$p < 0.001$ using (**A** and **F**) paired $t$-test and **C**–**E** one-way ANOVA with a Dunnett post hoc test. Assays were performed on **A**–**B** $n = 4$ (in duplicate), $n = 7$ (in quadruplet) and $n = 7$ for pancreatic, lung and skin tumor-bearing mice respectively, **C**–**D** $n = 8$, 4 and 8 (Pancreatic for basal media, FB CMed and CAF CMed treatment, respectively. All performed in triplicate) and $n = 9$, 6 and 9 (Lung for basal media, FB CMed and CAF CMed treated, respectively. Performed in duplicate for basal media and CAF CMed and only a single time for FB CMed treated), **E** $n = 3$ (in triplicate) and **F**–**G** $n = 8$ (in triplicate) independent experiments.

while the levels of fibrinogen were unchanged (Fig. 3E and Supplementary Fig. 4D) suggestive of a reduction in NET-mediated thrombosis within the circulation of treated mice.

Since the anti-tumor effects of t-NET inhibition were not directed by secondary effects on other infiltrating immune populations, we then examined the neutrophils themselves in more detail. Following treatment with CAF CMed and PAD4 inhibitor for 3 h in vitro to prevent NETs, neutrophils increased expression of activation markers CD11b and CD18, and completely shed surface CD62L (Fig. 3F), typical of activated neutrophils. Phagocytosis was reduced and intracellular ROS levels were exhausted, consistent with depletion following an oxidative burst (Fig. 3G, H). Furthermore, neutrophils exhibited enhanced degranulation with increasing expression of surface CD35 and CD63 after PAD4 inhibition that is associated with a pro-inflammatory phenotype (Fig. 3I). To test the pro-inflammatory, anti-tumor potential of degranulation following PAD4 inhibition, tumor cells were incubated in media from control or PAD4 inhibitor treated neutrophils that had been stimulated with CAF CMed. Media components from PAD4-inhibited neutrophils significantly impaired tumor cell growth compared to control conditions (Fig. 3J). Together these data support the concept that t-NETs are pro-tumorigenic, and their inhibition is sufficient to prevent growth of established tumors through the restoration of an inflammatory state in tumor-infiltrating neutrophils.

The success of PAD4 inhibition in melanoma was not recapitulated in the KPC genetic model of pancreatic cancer, with no significant difference in tumor size being observed after treatment (Supplementary Fig. 5A, B). Nonetheless, systemic effects were detected, with PAD4 inhibition reducing levels of clotting factors in the plasma (Supplementary Fig. 5C) as with melanoma. Similarly, no significant changes in the stromal landscape within the tumor were detected (Supplementary Fig. 5D, E). While the treatment was effective systemically, as evidenced by reduced clotting, we suggest that the lack of impact on these tumors, which still contained t-NETs (Fig. 1B), may be due to a lack of drug penetration which is a well-known phenomenon in mouse and human pancreatic cancer[39,40]. Indeed, supporting this, treatment of syngeneic pancreatic tumors (induced by injecting KPC-derived cells into wild-type mice; Supplementary Fig. 6A) with GSK484 completely suppressed tumor growth compared with vehicle-treated mice (Supplementary Fig. 6B) mirroring the effects seen in melanoma. Here, we observed that neutrophils and NETs (Supplementary Fig. 6C, D, respectively) were frequently confined to CAF-rich regions of vehicle-treated pancreatic tumors. As with melanoma, treatment

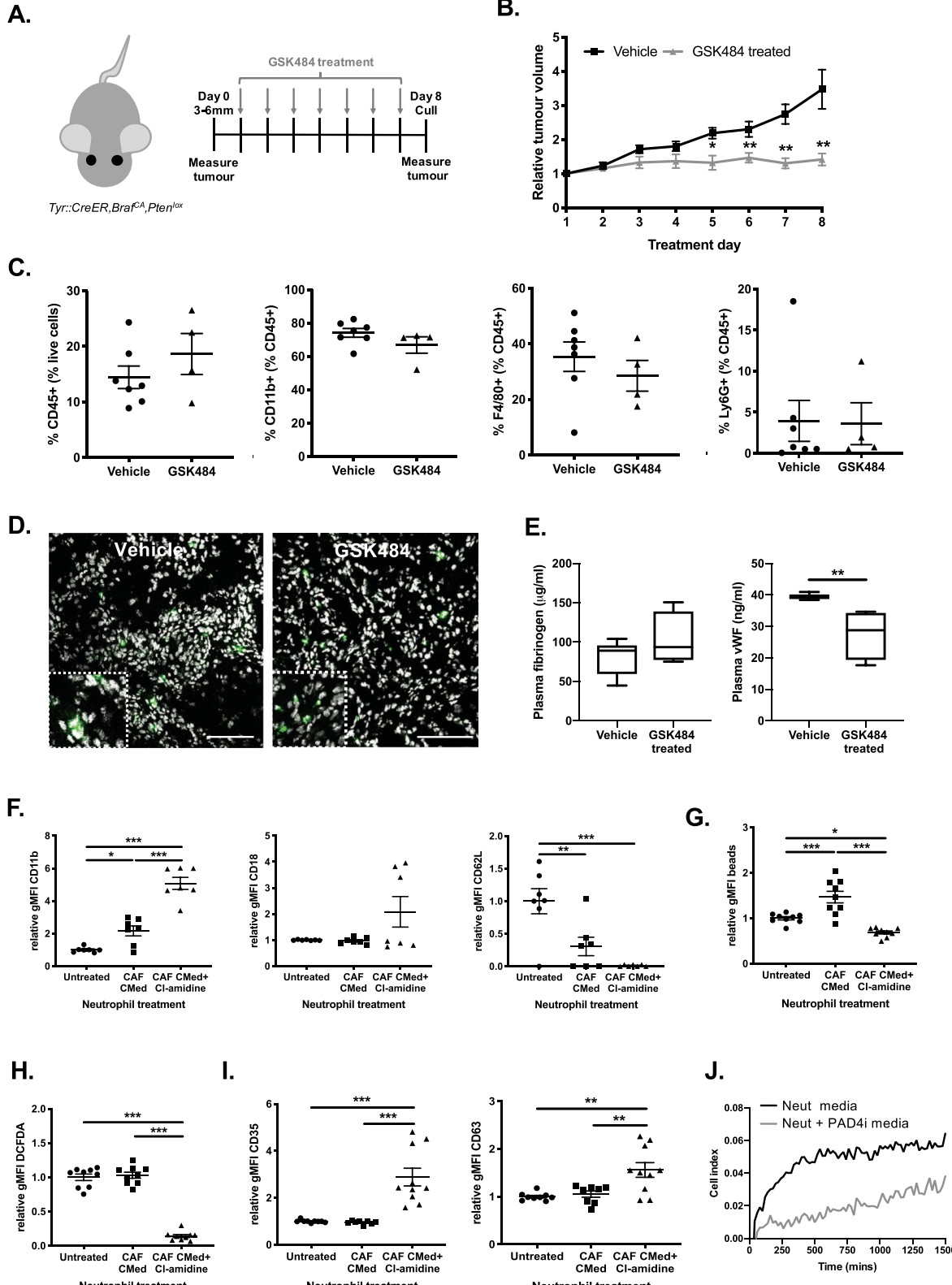

with GSK484 did not affect neutrophil infiltration into the tumor but NETs were absent (Supplementary Fig. 6C, D). Furthermore, GSK484 treatment suppressed the capacity of bone marrow neutrophils in tumor-bearing mice to NET, which coincided with a reduction in neutrophil death without affecting the number of neutrophils within the bone marrow (Supplementary Fig. 6E).

Moreover, in PAD4$^{-/-}$ mice, where neutrophils are genetically incapable of undergoing NETosis, induced pancreatic tumors grew significantly slower than those generated in wild-type littermates (Supplementary Fig. 6F), echoing the effects of pharmacologic NET inhibition. Thus, in multiple model systems, inhibiting PAD4 and prevention of NETosis exerted a significant

**Fig. 3 Inhibiting t-NETosis stops tumor growth in vivo. A** Schematic of GSK484 treatment regime of skin tumor-bearing mice. **B** Relative volume of skin tumors on mice treated with vehicle or GSK484 over 8d. **C** Flow cytometric analysis of the percentage immune cells (CD45$^+$), myeloid cells (CD11b$^+$), macrophages (F4/80$^+$) and neutrophils (Ly6G$^+$) in the tumor. **D** Representative confocal images showing neutrophils (myeloperoxidase (MPO) in green) within the tumor in control or treated animals (zoomed inset). Nuclei counterstained with DAPI (white). **E** The levels of clotting factors (fibrinogen and von Willebrand factor; vWF) in the plasma of skin tumor-bearing mice treated with vehicle or GSK484. **F** Quantification of CD11b, CD18 and CD62L expression on wild-type bone marrow-derived neutrophils after 3-h treatment with pancreatic CAF CMed, with and without Cl-amidine, in vitro by flow cytometry. **G** The phagocytic capacity of wild-type bone marrow-derived neutrophils after 3-h pancreatic CAF CMed, with and without Cl-amidine, treatment based on uptake of fluorescent 1 μm beads in vitro assessed by flow cytometry. **H** Quantification of ROS production by wild-type bone marrow-derived neutrophils after 3-h treatment with pancreatic CAF CMed, with and without Cl-amidine, in vitro based on the levels of DCFDA by flow cytometry. **I** Quantification of neutrophil degranulation based on CD35 and CD63 expression by flow cytometry after 3-h pancreatic CAF CMed, with and without Cl-amidine for 3 h. **J** Representative plot illustrating tumor cell growth following treatment with neutrophil-derived media with and without PAD4i treatment. Bar graphs are mean ± SEM; *$p < 0.05$, **$p < 0.01$ and ***$p < 0.001$ using **B** a Mann–Whitney test comparing vehicle and GSK484 at each time point, **E** a Mann–Whitney test and **F–I** a one-way ANOVA with a Tukey post hoc test. Box and whiskers graph-line: median, box: upper and lower quartiles, whiskers: maxima and minima. Assays were performed on male and female, 8–24-wk-old mice **B** $n = 8$ (Vehicle) and $n = 7$ (GSK484), **C** $n = 7$ (Vehicle) and $n = 4$ (GSK484), **D** Representative images of n-4 (Vehicle) and $n = 6$ (GSK484) tumors, **E** $n = 6$ (Vehicle) and $n = 4$ (GSK484) for Fibrinogen and $n = 7$ (Vehicle) and $n = 4$ for vWF, **F** $n = 3$ (in triplicate or single) **G** $n = 3$ (in triplicate) **H** $n = 3$ (in triplicate) **I** $n = 3$ (in triplicate) and **J** Representative of $n = 2$ (in duplicate) independent experiments. Scale bars are 100 μm.

---

effect on tumor growth. Collectively, these data indicate that t-NETs play a critical role in tumor progression.

**CAF-derived Amyloid β drives t-NET formation.** As neutrophils within the tumor traverse CAFs before entering the main tumor bulk, we next assessed whether direct contact was required for NET generation, or whether it was entirely soluble-mediator dependent as we observed systemically. To evaluate the capacity of CAFs to induce NETs by direct cell-cell interaction vs. secreted factors, neutrophils were seeded onto pancreatic tumor-derived CAFs in the presence or absence of FB or CAF CMed. While CAFs in the presence of CAF CMed readily induced NETosis, CAFs with normal media or FB CMed could not (Fig. 4A). This suggested that direct contact between CAFs and neutrophils was not sufficient to drive t-NETosis, and the effect is primarily mediated through factors secreted by the CAFs. We established that the CAF-derived factor was not vesicle bound as depletion of microvesicles in the CMed had no effect on CAF CMed driven NETosis (Fig. 4B), and media fractionation further confirmed that the soluble mediator driving NETosis was a protein and not a metabolite (Fig. 4C).

Therefore, we performed proteomic analysis of the pancreatic FB and CAF CMed to identify the potential t-NET driver(s). Mass spectrometry revealed a number of differentially secreted proteins (Fig. 4D and Supplementary Fig. 7) between these two cell types. Of interest, Amyloid β A4 protein (APP), fibronectin and heat shock protein 90 were significantly upregulated in CAF CMed (Fig. 4E), all of which have been implicated in NETosis in different disease contexts[41–44]. Treatment of neutrophils with a fibronectin inhibitor had no effect on the ability of CAF CMed to induce NETs (Supplementary Fig. 8A), thus, we examined APP in greater detail. Similar to studies implicating Amyloid β peptide driven NETosis in Alzheimer's[45], *app* mRNA was detected in pancreatic CAFs at higher levels than pancreatic stellate cells (Supplementary Fig. 8B) and this was maintained at the protein level with Amyloid β detected in CAF CMed (Fig. 4F). Inhibition of β-secretases, which regulate secretion of Amyloid β (BACE 1–2), by pancreatic CAFs in turn reduced Amyloid β in the media to levels comparable with basal controls (Fig. 4F). Upon determining that recombinant Amyloid β but not Amyloid α was able to induce NETosis in a dose-dependent manner (Supplementary Fig. 8C) we next assessed whether Amyloid β was present in NET-rich tumors. In pancreatic tumors, two distinct patterns of APP distribution were observed with regions displaying either diffuse or punctate staining that co-localized

with CAFs and tumor cells (Fig. 4G) consistent with Amyloid β aggregates observed in other diseases. These data suggest that CAFs are a source of Amyloid β in the tumor, a key driver of NETosis.

To confirm that it was Amyloid β within CAF CMed that was responsible for the observed effects on neutrophils in tumors, we inhibited BACE 1–2 in CAFs to prevent Amyloid β production (as shown in Fig. 4F). BACE inhibition in pancreatic and lung CAFs abolished the ability of the CMed to induce NETs in vitro (Fig. 4H) supporting the hypothesis that Amyloid β production by CAFs underlies their capacity to induce t-NETs. A trend towards reduced neutrophil death was also observed in the absence of Amyloid β (Supplementary Fig. 9A) indicating that ROS-mediated, suicidal NETosis in the tumor context is potentially driven by a single CAF-derived factor.

To further support our observations, I/V infusion of CAF CMed derived from CAFs pre-treated with BACE inhibitor to suppress Amyloid β production, abolished the systemic effects measured with CAF CMed (Fig. 4I and Supplementary Fig. 9B), whilst the effect of CAF CMed in vivo was recapitulated by infusion of recombinant Amyloid β alone (Fig. 4I). We next examined the effects of BACE inhibition in vivo, in skin tumor-bearing mice (Fig. 4J) using the same treatment regime as Cl-amidine and GSK484. As with inhibitors of NETosis, disruption of Amyloid β secretion prevented further increases in tumor volume compared with vehicle controls (Fig. 4J), implying that CAF-driven Amyloid β-mediated NETosis supports tumor development in vivo, and that perturbation of either the NET driver or NET process is sufficient to prevent growth. This was confirmed using the B16.F10 melanoma model which is both neutrophil and NET poor[35]. Treatment with CAF CMed or recombinant Amyloid β increased tumor growth (Fig. 4K), potentially coinciding with enhanced infiltration of NETting neutrophils (Supplementary Fig. 9D).

We then considered how CAF-derived Amyloid β may exert its effects on neutrophils. In light of reports that CD11b may act as a receptor for Amyloid β[46], we blocked CD11b during CAF CMed treatment and found that it completely abolished NETosis (Fig. 4L and Supplementary Fig. 9C) without affecting other functions such as phagocytosis (data not shown). In contrast, TLR2 neutralization on CAF CMed treated neutrophils had no effect on their capacity to NET (Fig. 4M and Supplementary Fig. 9C). Thus, CD11b on the surface of neutrophils may indeed function as a receptor, with its blockage desensitizing cells to the effects of Amyloid β present in CAF CMed.

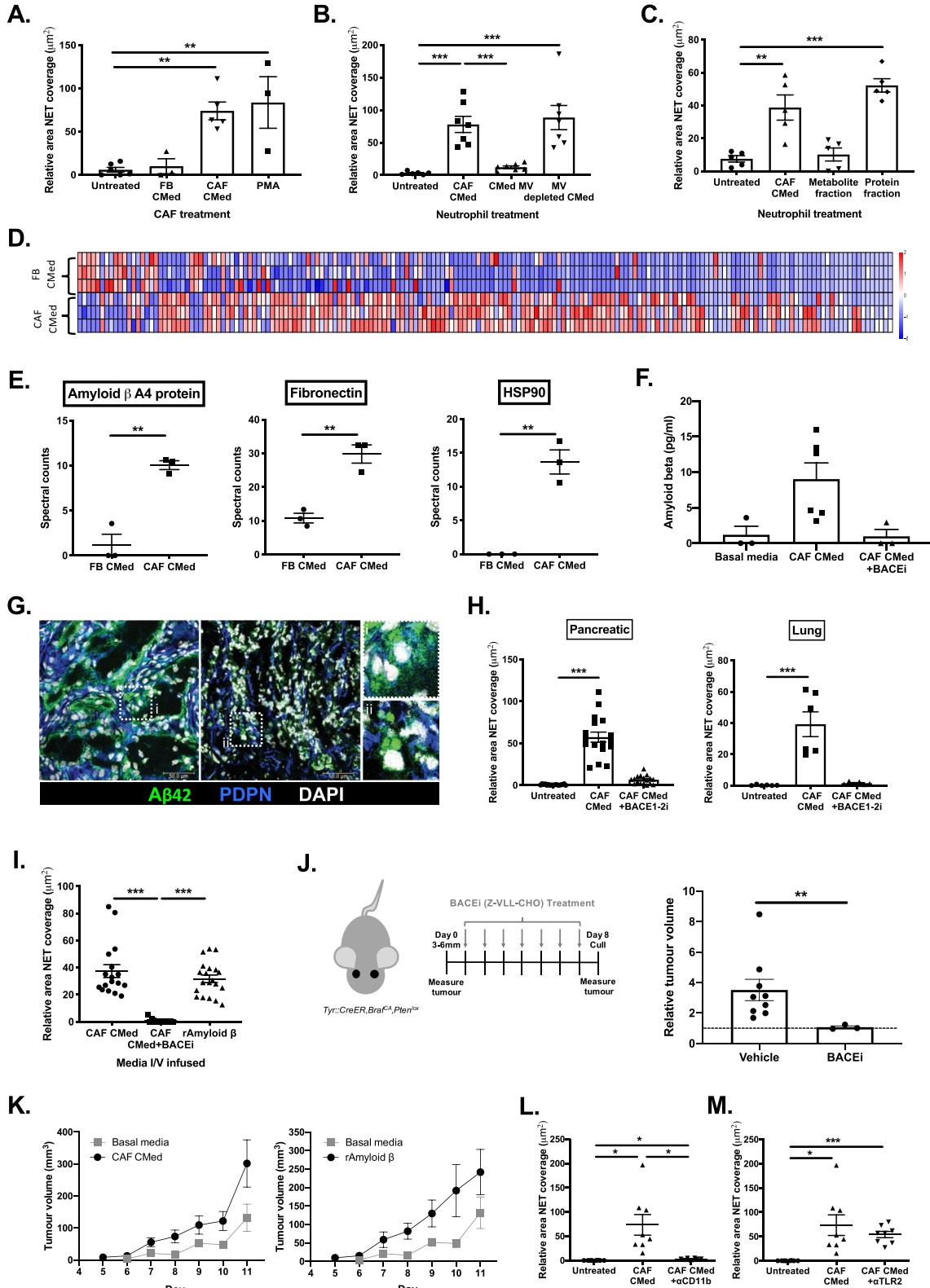

**NETs reciprocally activate CAFs**. Whilst we observed that t-NETosis was entirely soluble mediator driven (Fig. 4A), the fact that NETs observed in the primary tumor site were often restricted to CAF dense regions (Fig. 1A) led us to ask if t-NETs had reciprocal, pro-tumor effects on the phenotype and function of CAFs in their proximity. Treatment of CAFs with NETs

supported an enhanced proliferation of CAFs (Fig. 5A), and induced features of activation in vitro (Fig. 5B–E).

We observed an increase in contraction of CAFs based on the appearance of large gaps between cells after treatment with NETs derived from CAF CMed treated neutrophils compared to FBs treated with NETs, which remained as a monolayer (Fig. 5B). To

**Fig. 4 Amyloid β is the driver of CAF-induced t-NETosis. A** Quantification of the relative NET coverage of neutrophils that were added to lung CAFs treated with or without FB CMed, CAF CMed or PMA for 3 h. **B** Quantification of the relative NET coverage of neutrophils treated with lung CAF CMed or CAF CMed-derived microvesicles (MV) or CMed depleted of MV. **C** Quantification of the relative NET coverage of neutrophils treated with lung CAF CMed or the metabolite or protein fractions of the CAF CMed. **D** Differentially secreted proteins in pancreatic FB and CAF CMed analyzed by mass spectrometry. The relative intensity of each protein is indicated by the colored bar under the heatmap. **E** Spectral counts for NET-related factors secreted by pancreatic FBs and CAFs. **F** Levels of Amyloid β in basal media and pancreatic CAF CMed generated in the presence or absence of an inhibitor of amyloid-beta secretion (BACEi). **G** Confocal microscopy of amyloid precursor protein (APP) and Podoplanin (PDPN) expressed by CAFs in pancreatic tumors. Insets depict diffuse vs. aggregated patterns of APP/Amyloid β distribution. **H** Quantification of the relative NET coverage of neutrophils treated with pancreatic and lung CAF CMed generated with or without 24-h pre-treatment of the CAFs with a β-secretase 1 and 2 (BACE1-2) inhibitor. **I** Quantification of the relative NET coverage of bone marrow neutrophils taken from wild-type mice intravenously infused with pancreatic CAF CMed taken from cells treated with or without BACE1-2 inhibitor for 24 h or recombinant Amyloid β. **J** Schematic of BACEi treatment regime of skin tumor-bearing mice and relative endpoint volume of skin tumors (calculated relative to the volume of each tumor at the start of treatment—indicated by dotted line on the graph) on mice after treatment with vehicle or BACEi (Z-VLL-CHO). **K** Growth of orthotopically implanted B16.F10 tumor cells with vehicle, CAF CMed or recombinant Amyloid β treatment. Quantification of the relative NET coverage after 3-h treatment with pancreatic CAF CMed with or without **L** CD11b or **M** TLR2 blocking antibodies. Data are mean ± SEM; **$p < 0.01$ and ***$p < 0.001$ using (**A** and **H**) one-way ANOVA with a Dunnett post hoc test, **B–C**, **I** and **L–M** one-way ANOVA with a Tukey post hoc test and **J** a Mann–Whitney test. Assays were performed on **A** $n = 6$ (Untreated and CAF CMed treated) $n = 3$ (PMA and FB CMed treated), **B** $n = 7$, **C** $n = 5$, (**E**) $n = 3$, **F** $n = 3–6$, **G** Representative images of $n = 3$ tumors, **H** $n = 6$ (in triplicate), **I** $n = 7$ (in duplicate or triplicate), **J** $n = 9$ (Vehicle) and $n = 3$ (BACEi) male and female, 8–24-wk-old mice, **K** $n = 6$ 8-wk-old female C57BL/6 mice and **L–M** $n = 3$ (in triplicate) independent experiments. Scale bars are 50 μm.

assess the contractile properties of the CAFs after treatment with t-NETs, we performed contraction assays. t-NET treated CAFs contracted collagen gels to a greater extent than CAFs treated with unstimulated NETs (Fig. 5C). This coincided with an increased expression of αSMA and Col1a2 at the RNA level (Fig. 5D) and collagen I imaging indicated these changes may be reflected at the protein level (Fig. 5E). Examination of skin tumors treated with either PAD4 or BACE inhibitors revealed a reduced cellularity per unit tumor volume when compared with vehicle-treated tumors (Fig. 5F, G). Moreover, consistent with earlier observations that NET inhibition promoted degranulation and impaired tumor cell growth (Fig. 3I, J), a similar effect was mirrored in CAFs (Fig. 5H). Fibrotic structures with more mature collagen made up the remaining tumor mass (Fig. 5G lower panel). Collectively, these data suggest that neutrophils skewed to undergo NETosis likely localise to CAF-rich regions of a tumor where they form t-NETs in response to Amyloid β, promoting CAF expansion, contractility and deposition of matrix components supportive of tumor growth.

**Conservation of Amyloid β-NET axis in human disease.** Having observed a significant pro-tumor communication between CAFs and neutrophils in multiple murine models, we sought to determine the clinical significance of these findings by examining human tumors. t-NETs were observed in human pancreatic tumors (Fig. 6A), melanoma and melanoma metastases (Fig. 6B and Supplementary Fig. 10A), and when detected, both NETs and neutrophils were found juxtaposed to CAFs mirroring murine tumors. We next examined publicly available datasets for APP and β-secretase expression. Significantly, both pancreatic adenocarcinoma and cutaneous melanoma expressed elevated levels of APP and BACE2 compared to matched normal tissue (Fig. 6C, D). When detected, elevated levels of Amyloid β could be measured circulating in the blood of patients with advanced melanoma compared with heathy controls (Supplementary Fig. 10B). Moreover, expression of both *app* and *bace2* strongly correlated with stromal markers *pdpn, acta2, col1a2, cd34* typically used to identify CAFs, but not the lymphatic marker *lyve-1* (Fig. 6E) inferring that correlations were specific to CAFs and not other components of the tumor stroma. These data recapitulate murine tumors, supporting CAFs as a source of Amyloid β in human pancreatic cancer and melanoma. The high expression of *bace2*, the rate-limiting step in Amyloid β release, was correlated with poorer patient prognosis in both tumor types (Fig. 6F) while high

expression of *app* did not correlate with patient outcome (Fig. 6G). Collectively, these findings suggest that Amyloid β secreted by CAFs is a critical driver of t-NET formation conserved in human cancers associated with poor prognosis, and importantly, has the potential to be detected in liquid biopsies.

## Discussion
The stroma is critical to tumor development and progression and it is evident that targeting stromal interactions offers many opportunities for cancer treatments[1]. Here, we report the existence of a previously undescribed pro-tumor crosstalk between non-immune and immune stromal components to support tumor growth. We demonstrate that neutrophils recruited to the tumor frequently localize to CAF-rich areas, where they are stimulated to generate extracellular traps (NETs) supporting tumor growth. This process, which we have termed tumor-induced NETosis (t-NETosis), is driven by CAF-secreted Amyloid β both locally via CD11b on neutrophils within the primary tumor and at the systemic level within the bone marrow niche.

While neutrophil function has been extensively studied in the context of inflammation and tissue damage[47–49], there have been fewer studies relating to their contribution to the primary tumor[32,50,51] and mediators responsible for driving their release[21,51]. Contradictory studies have reported NET-derived proteins acting to both promote tumor cell proliferation and invasion[52,53], as well as inhibit tumor growth through apoptosis induction[32,54]. In advanced disease, NETs have been reported to contribute to thrombus formation[21], metastatic colonization[22,23] and most recently the reactivation of dormant tumor cells[29]. Previous studies point to a central role for G-CSF in recruiting neutrophils and to date this is one of the few factors that has been identified as a NET inducer[21,35]. This is particularly thought provoking given that cancer patients that are at high risk of becoming neutropenic receive G-CSF therapy in an attempt to restart myelopoiesis after chemo- and radiotherapy[55]. Consistent with previous studies we found that CAF-induced NETosis was ROS and PAD4-dependent[14], but in this case did not require G-CSF.

With CAFs possessing such a potent effect on neutrophils via mechanisms distinct from previously described for systemic NETosis[21], we then examined the contribution of NETs in tumors. Inhibition of t-NETosis in mice with established tumors had no impact on neutrophil infiltration, yet was sufficient to abolish further growth. In these tumors, PAD4 inhibition

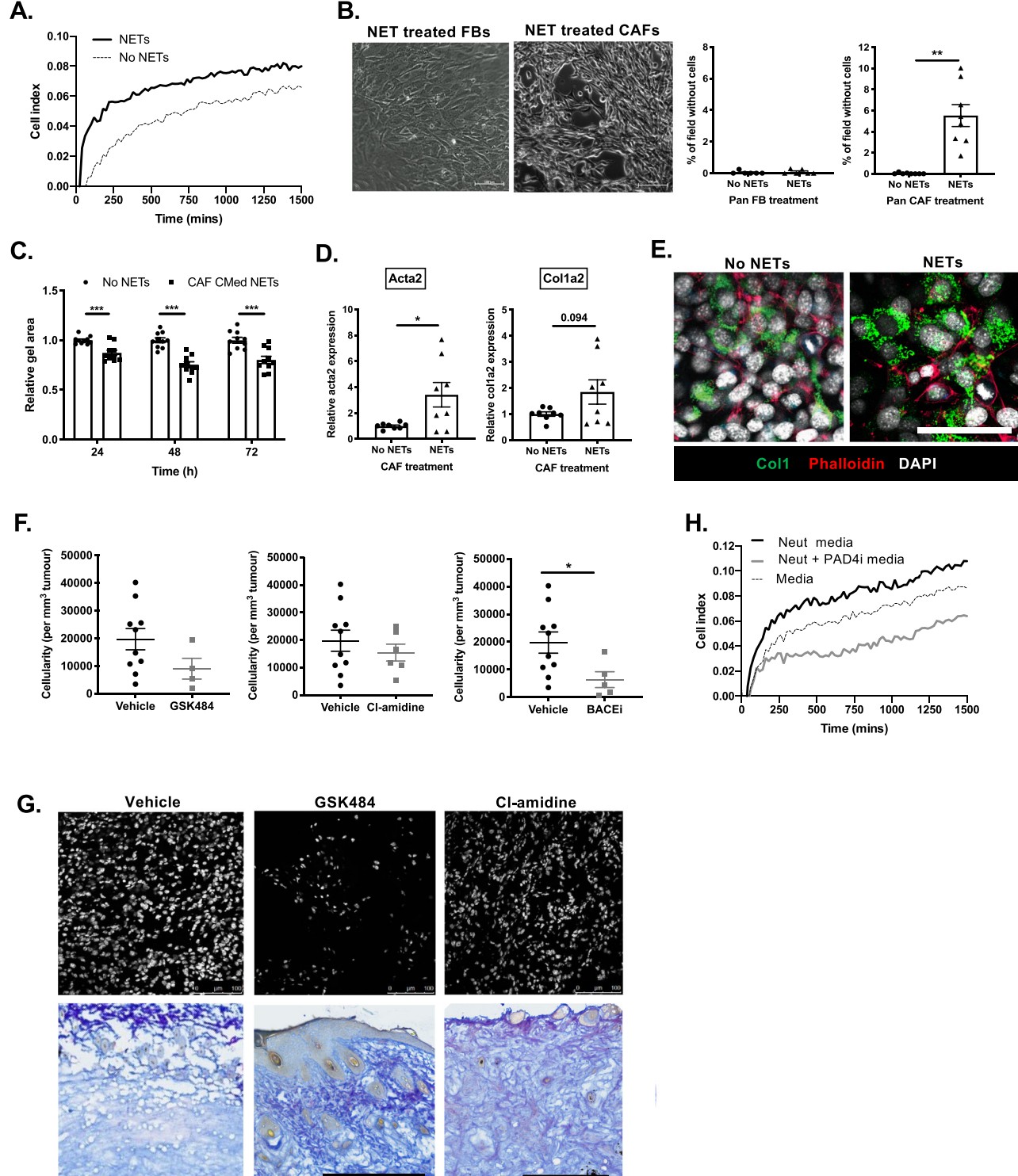

translated to a skewing of neutrophils to a cytotoxic, anti-tumor N1-like phenotype[19,56]. Indeed, our data supported this hypothesis, confirming enhanced expression of activation markers and degranulation after PAD4 inhibition.

We subsequently identified Amyloid β, a protein typically associated with neurodegenerative disorders, as the critical CAF-derived factor inducing t-NETs. Inhibition of Amyloid β or its proposed cognate receptor, CD11b[46], prevented CAF-mediated NETosis and reduced growth of established tumors. Moreover, the addition of soluble Amyloid β exacerbated tumor growth

further implicating NETosis as pathological response in the cancer context. With the observed increase in expression of CD11b on neutrophils after CAF CMed treatment, our data imply that CAFs not only secrete the Amyloid β to drive NETosis, but may also render the neutrophils more responsive to circulating Amyloid β in tumor-bearing animals through increased expression of its receptor[46]. This is in line with reports that Amyloid β can induce ICAM-1 expression by endothelial cells suggesting that it may play additional roles in the high-affinity capture of infiltrating neutrophils[41,57].

**Fig. 5 t-NETs induce CAF activation. A** Representative plot illustrating growth of CAFs in the presence of vehicle or micrococcal nuclease detached NETs. **B** Phase contrast images and respective quantification of the percentage area of cell free space per field after treatment of pancreatic FBs or CAFs with NETs derived from CAF CMed treated neutrophils for 24 h. (**C**) Quantification of the size of collagen gels 24, 48, and 72 h after seeding pancreatic CAFs treated with or without NETs derived from CAF CMed treated neutrophils for 24 h. **D** Expression of *Acta2* and *Col1a2* at the gene level in pancreatic CAFs treated with or without NETs derived from CAF CMed treated neutrophils for 24 h. **E** Confocal microscopy of Collagen1 and phalloidin on pancreatic CAFs treated with or without NETs derived from unstimulated neutrophils or CAF CMed treated neutrophils for 24 h. **F** Cellularity of melanoma after treatment with GSK484, Cl-amidine, BACEi or vehicle expressed as cells per unit volume. **G** Representative confocal image of nuclei and collagen after treatment with GSK484, Cl-amidine, or vehicle. Tissues stained with Herovici stain to demark new collagen (blue) vs. mature collagen (pink). **H** Representative plot illustrating growth of CAFs following treatment with CAF CMed or CAF CMed and PAD4i-treated neutrophil-derived media. Bar graphs are mean ± SEM; $*p < 0.05$ and $***p < 0.001$ using **B–D** a paired *t*-test and **F** a Mann–Whitney test. Box and whiskers graph-line: median, box: upper and lower quartiles, whiskers: maxima and minima. Assays were performed on (**A** and **H**) Representative of $n = 2$ (in duplicate), **B** $n = 3$ (in triplicate), **C** $n = 3$ (in duplicate or quadruple), **D** $n = 4$ (in duplicate), **E** Representative images of $n = 3$ samples, **F** $n = 5$–10 male and female, 8–24-wk-old mice, and **G** Representative images of $n = 5$ (Vehicle) and $n = 5$ (GSK484 and Cl-amidine) independent experiments. Scale bars are 50 μm (**B** and **E**), 500 μm (**G** upper panel) and 100 μm (**G** upper panel).

In Alzheimer's disease, Amyloid-driven NETosis has been associated with poor prognosis as a consequence of endothelial and parenchymal damage, and neurotoxicity[57–59]. In this context, aggregation of Amyloid β not only drives the cognitive symptoms associated with Alzheimer's, but its soluble form also acts as a DAMP, contributing to the neuroinflammation which perpetuates progression[60,61]. In cancer, we observed that Amyloid β secreted by CAFs form microaggregates in the tumor to drive NETosis in situ as well as disseminating into the blood where it conditions circulating and bone marrow resident neutrophils. This raises the possibility that microaggregates of CAF-derived Amyloid β also act as a DAMP inducing aberrant NETosis to support tumor progression.

We provide evidence that neutrophils in pancreatic, skin and lung cancers exposed to CAF-derived cues exert pro-tumor effects that operate on multiple levels. Suppression of t-NETs in tumor-bearing mice by inhibition of PAD4, or release of Amyloid β by BACE inhibition not only suppressed tumor growth but also brought about a decrease in thrombus formation as quantified by levels of clotting factors; vWF and fibrinogen. With previous reports showing that NETs both contribute to thrombus formation, metastatic colonization and activation of dormant tumor cells[21–23,29,35] our findings that perturbation of either the driver or consequence of NETosis is sufficient to prevent tumor growth present the CAF-Amyloid-neutrophil axis as an attractive target. While the therapeutic potential of PAD4 blockade has yet to be clinically tested, it is possible that this approach may only provide a narrow therapeutic window since the inhibitor acts when PAD4 is in a high calcium-binding state[62], and may increase susceptibility to infection where NETosis is a critical response for bacterial clearance. Thus, inhibiting Amyloid β, for which drugs targeting BACE's have already received FDA approval[63], and preventing neutrophils from receiving a NET stimulus may represent a more effective platform. Importantly, this would block pathological NETosis both at the primary tumor and systemically without impacting other critical neutrophil functions.

With the growing body of evidence supporting a role for NETs in advanced cancer—metastatic colonization and recurrence[23,29,64], and clinical data linking neutrophil infiltration with poor prognosis in multiple cancer types[65–69]—our findings warrant further studies to determine whether Amyloid β (circulating, or in the primary tumor) can be used as a biomarker to stratify patients. Although, in neuroinflammatory conditions such as Alzheimer's disease, circulating Amyloid β is not currently considered to be a biomarker reflective of levels in the cerebrospinal fluid and central nervous system[70], we reported no significant difference in Amyloid β levels in our small cohort of melanoma patients compared to healthy volunteers. However, this would merit assessment in a much larger cohort. A number of external factors may be responsible for the observed variability; Circulating Amyloid $\beta_{1-40}$ and $\beta_{1-42}$ are predominantly bound to plasma proteins, and platelets or erythrocytes which can mask detection[70]. Furthermore, differences in the hepatic and renal clearance of Amyloid β may influence the levels measured[70]. Despite the difficulties in quantifying plasma Amyloid β levels, recent studies have implicated plasma Amyloid β in cancer. Higher levels of plasma Amyloid β were detected in hepatic cancer patients than those with Alzheimer's disease[71], indicating that while cerebrospinal fluid levels are predictive of Alzheimer's, circulating Amyloid β levels may be more reflective of cancer[71]. Furthermore, it has been reported that breast and prostate tumor cells undergo amyloidosis as a result of environmental stress leading to cancer cell dormancy. This is particularly interesting in light of the recent study which implicated NETosis as a driver of dormant cancer cell activation in lung cancer[72]. Indeed, amyloid accumulation in tumor cells may force the cells into a dormant state but as a consequence may also induce NETosis which could result in their activation and in a negative feedback system, re-initiation of the tumor.

Amyloid β-induced NETosis potentially support tumor growth through reciprocal effects on CAFs in close proximity. Here we showed that t-NETs supported changes in morphology, proliferation and activation status of CAFs leading to a more contractile and matrix-producing phenotype. Activation, matrix deposition and consequent matrix stiffening are all pro-tumor traits exhibited by CAFs that are correlated with disease progression and promotion of tumor growth in many cancers[1,4]. Without directly affecting tumor cell proliferation, alterations to CAF phenotype and functional traits have the potential to impact multiple components of the tumor microenvironment via several mechanisms, including the direct support of cancer cell invasion, growth factor release, cytokine production and bioavailability, as well as, modulating functional status, physical exclusion, and altered trafficking of immune cells. Similar phenomena have also been reported in fibrosis where NETosis was found to drive activation, differentiation and the fibrotic response of human lung fibroblasts[18]. Therefore, the direct action of Amyloid β on tumor cells previously reported[72] is distinct to the indirect effects it has on the tumor stroma through the induction of NETosis, further supporting a potential rationale for using BACE inhibitors to treat cancer.

In summary, we have demonstrated the existence of a novel mechanism by which CAFs stimulate t-NETosis at local and systemic levels via production of Amyloid β. Targeting this process with existing small molecule inhibitors stops tumor growth and restores a pro-inflammatory neutrophil phenotype. This sets the stage for further studies examining the potential of therapeutically targeting NETs to treat cancer.

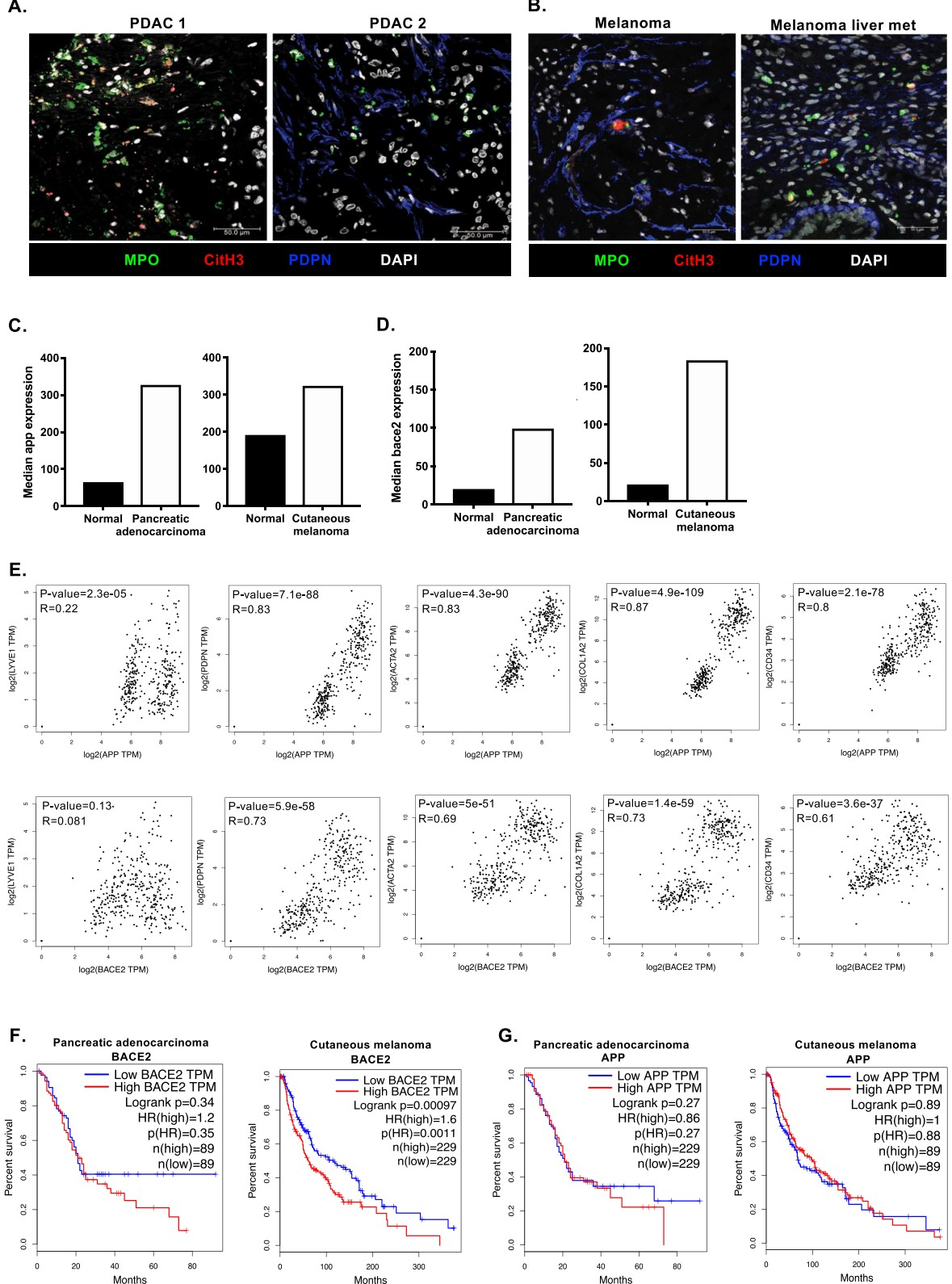

## Methods

**Mice**. The care and use of all mice in this study were in accordance with the UK Animals in Science Regulation Unit's Code of Practice for the Housing and Care of Animals Bred, Supplied or Used for Scientific Purposes, the Animals (Scientific Procedures) Act 1986 Amendment Regulations 2012. All procedures were performed under a UK Home Office Project license (PPL 80/2574 or PPL P88378353), which were reviewed and approved by the Medical research Council Laboratory of Molecular Biology (PPL 80/2574 or PPL P8837835) Animal Welfare and Ethical Review

Bodies (AWERB). Mice were kept in specific pathogen-free (SPF) barrier conditions and once recruited to studies, remained socially housed in individually ventilated cages, at ambient temperature and with cage enrichment. For spontaneous genetic mouse tumors, Tyr::CreER; Braf$^{CA}$; Pten$^{lox/lox}$ (melanoma; stock number 013590, The Jackson Laboratory), the LSL-KrasG12D/+;LSLTp53R172H/+;Pdx-1-Cre (pancreatic tumor, provided by Tobias Janowitz), and inducible LSLKrasG12D/+;p53LSL-R270H/ER (lung tumor, as described in ref. [73]) model systems were tested. Orthotopic syngeneic tumors using B16.F10 melanoma were performed in ~8-week-old female

**Fig. 6 Conservation of t-NETs in human disease. A** Representative confocal images of NETting neutrophils and CAFs in human pancreatic tumor biopsies (MPO, green; CitH3, red; podoplanin, blue; DAPI, white). **B** Representative confocal images of NETting neutrophils and CAFs in human melanoma primary tumor and metastasis samples (MPO, green; CitH3, red; podoplanin, blue; DAPI, white). **C** Median expression of *app* in human pancreatic adenocarcinoma and cutaneous melanoma compared to normal tissue from healthy donors and **D** Median expression of *bace2* in human pancreatic adenocarcinoma and cutaneous melanoma compared to normal tissue from healthy donors (GTEx and TCGA datasets). **E** Correlation of *app* and *bace2* with a lymphatic marker (*lyve1*) and CAF markers (*pdpn, acta2, col1a2* and *cd34*). Kaplin–Meier curves showing overall survival of pancreatic adenocarcinoma and cutaneous melanoma patients correlated with high and low expression of **(F)** *bace2* or **(G)** *app*. **(A)** Representative images of at least n = 3 tumors. Scale bars are 50 μm.

C57BL/6 mice bred in house. Syngeneic subcutaneous pancreatic tumors (cells provided by Professor David Tuveson, CSHL) were generated in ~8-week-old female C57BL/6 mice or PAD4−/− (provided by Markus Hoffmann, University of Erlangen and originally derived from Kerri Mowen, Scripps Institute) and littermate control mice (on a C57BL/6 genetic background).

**PAD4 inhibition in vivo.** Male and female spontaneous skin (8–24-wk old) or pancreatic (8–32-wk old) tumor-bearing mice, were treated with PAD4 inhibitors; 3.5 mM Cl-amidine (EMD Millipore), 20 mg/kg GSK484 (Cayman Chemicals) or a vehicle control (DMSO). Where possible, technicians performing the experiment were blinded to drug treatments. Mice were recruited when tumors reached between 3–6 mm in diameter and then received I/P doses of PAD4 inhibitor or vehicle control every day for 7d or until tumors reached their size limit (12 mm max. diameter) and mice were then euthanized by exposure to $CO_2$ followed by cervical dislocation or exsanguination as a confirmation of death. The size of genetic pancreatic tumors was monitored by high-resolution ultrasound, as previously described[2]. For skin and subcutaneous pancreatic tumors, the volume was recorded daily using the formula (π/6)(shortest length*longest length)[2]. All mice in the study were monitored daily, and tumor size measured non-invasively with digital calipers. After treatment, the cellularity of the tumors was determined by flow cytometric analysis of the number of cells present in the tumor after digestion of the tissue. The number of cells was then normalized to tumor size. Plasma was isolated from skin and pancreatic tumor-bearing mice by collecting the blood from cardiac puncture and centrifuging at $800 \times g$ for 10 min. Plasma was then snap frozen for measurements of analytes at a later stage.

**BACE inhibition in vivo.** Spontaneous skin tumor-bearing (male and female, 8–24-wk old) mice were treated with 5 mg/kg BACE inhibitor (Z-VLL-CHO, Abcam) or vehicle control. Mice were recruited when tumors reached between 3–6 mm in diameter and then received I/P doses of BACE inhibitor for 7d or until tumors reached their size limit (12 mm max. diameter) and mice were then euthanized by exposure to $CO_2$ followed by cervical dislocation or exsanguination as a confirmation of death. Mice were monitored and measured daily with digital calipers, and tumor volume was calculated daily using the formula (π/6) (shortest length*longest length)[2]. After treatment, the cellularity of the tumors was determined by flow cytometric analysis of the number of cells present in the tumor after digestion of the tissue. The number of cells was then normalized to tumor size.

**Amyloid β and CAF CMed treatment of tumors.** $2.5 \times 10^5$ B16.F10 melanoma cells were inoculated subcutaneously into the shoulder region of 8-week-old female C57BL/6 mice. Where possible, technicians performing the experiment were blinded to drug treatments; At day 5, day 7, and day 9, mice received I/P infusion of either vehicle, CAF CMed or recombinant Amyloid β. Mice were monitored and measured daily with digital calipers, and tumor volume was calculated using the formula (π/6)(shortest length*longest length)[2]. After 11 days or when the tumors reached the size limit, the mice were euthanized. The immune landscape and CAF composition of the tumors was analysed by flow cytometry.

**CAF CMed treatment in vivo.** For investigations of CAF factors in vivo in the absence of tumors, CAF or FB CMed diluted 1:1 in complete culture media or 250 μg/ml recombinant amyloid β was I/V infused into C57BL/6 mice and the bone marrow was harvested 24 h later.

**Cell isolation and culture.** CAFs were isolated from skin, lung (58) and pancreatic tumor-bearing mice by mechanical separation of the tumor followed by digestion with an enzymatic cocktail consisting of 1 mg/ml collagenase A and collagenase D and 0.4 mg/ml DNase I (all from Roche) in PBS at 37 °C for 2–3 h with rotation at 600 rpm in a thermomixer compact (Eppendorf). 10 mM EDTA was then added to stop the enzymatic reaction. Normal FBs were isolated from matched tissues of wild-type mice using the same method. CAFs and normal FBs were maintained in RPMI (Sigma- Aldrich) with 1.5 g/L NaHCO3, 10% fetal bovine serum (Life Technologies), 1% penicillin-streptomycin (Sigma-Aldrich), 10 mM HEPES (Gibco), 15 μM β-mercaptoethanol (Sigma-Aldrich).

**CMed generation and fractionation.** CMed from FB and CAFs was generated by culturing the cells until they reached 40–50% confluence and then changing the media to complete endothelial cell culture medium (Generon). The media was then harvested after 24 h and filtered through 0.2 μm cell strainers before freezing. In some experiments, pancreatic or lung CAFs were treated with an inhibitor to beta-site Amyloid precursor protein cleaving enzyme 1 and 2 (BACE) (0.7 μM; Abcam) for the duration of CAF CMed generation to inhibit Amyloid β production by the cells.

CAF CMed was also separated into the metabolite (<3 kDa) and protein (more than 3 kDa) fractions using 3 kDa centrifugal filters (Merck Millipore) according to the manufacturer's instructions. To isolate extracellular vesicles (MVs), CAF CMed was ultracentrifuged at 100,000 g for 90 min. MV depleted CMed was collected and the isolated MVs were resuspended in complete culture media.

**Neutrophil isolation.** Wild-type male and female C57BL/6 mice were euthanized by cervical dislocation. Femurs and tibias were removed and bone marrow aspirate was collected. Neutrophils were isolated using a two-step Histopaque density gradient as previously described[74]. Purified neutrophils were washed in PBS. To test the NETting capability of neutrophils from tumor-bearing mice, bone marrow from mice bearing skin, lung and pancreatic tumors of varying size were isolated.

**NETosis assay.** Bone marrow-derived neutrophils were counted and seeded onto poly-D-lysine coated plates at a density of $1 \times 10^5$ cells per condition in serum-free media. The media was changed to complete endothelial cell culture medium (Generon). To study NETosis, neutrophils were either treated with 1 μg/ml PMA, CMed derived from FB or CAFs mixed 1:1 with complete culture media. Neutrophils were treated for 3 h at 37 °C and 5% $CO_2$ and then stained with 20uM SYTOX™ Green Nucleic Acid Stain for 10 min (Thermofisher Scientific). Images and videos were taken using a Zeiss Axio Observer.Z1 coupled with incubation chamber (ZEISS). Five images were taken per well and each condition was performed in duplicate or triplicate.

To test the effects of ROS on NETosis, neutrophils were treated with anti-oxidants; 10 mM n-acetyl cysteine (NAC), 20 μM DPI, 2 mM Trolox and 2 mM Vitamin C for 30 min prior to inducing NETosis and for the duration of the NET assay. Alternatively, neutrophils were treated with 0.03 μg/ml α-G-CSF (R&D Systems) or 50 μM Chloroquine (Sigma-Aldrich) for the duration of the NET assay. In some experiments, neutrophils were treated with 1.5 mg/ml Cl-amidine, an inhibitor of the PAD4, 10 μg/ml anti-CD11b or anti-TLR2 (Biolegend) or 1 μg/ml fibronectin inhibitor (Santa-Cruz Biotechnology) for the duration of culture. The area of NET coverage was quantified using ZEN Lite (ZEISS) and ImageJ (Fiji) software by drawing around every NET within a field. Dead cells (positive for SYTOX green) were also counted per field.

**NET isolation and quantification.** After NET generation, neutrophils were treated with 1U/ml micrococcal nuclease (Sigma-Aldrich) for 10 min (found to be the optimum time for detaching NETs from neutrophils without digesting the NETting DNA) at 37 °C and 5% $CO_2$. The enzyme was inactivated with 0.5 mM EDTA.

**Gel contraction assays.** Pancreatic CAFs were treated for 24 h with ~10 ng/ml NETting DNA (taken from $1 \times 10^5$ neutrophils stimulated with CAF CMed or PMA) or from untreated neutrophils. CAFs were trypsinized and seeded into 2 mg/ml collagen gels (Rat tail collagen, BD Biosciences) at a density of $1 \times 10^5$ cells/gel in 24-well plates. Gels were left to polymerise for 20 min at 37 °C before adding full media. The gel was detached from the culture dish using a pipette tip. The gels were imaged at 24, 48, and 72 h after generation. The area of the gel was then measured using ImageJ software. The relative gel area was then calculated by comparing it to gels containing untreated CAFs.

**Flow cytometry.** Tumors were minced using a razor and digested with 1 mg/ml collagenase A and collagenase D and 0.4 mg/ml DNase I in PBS at 37 °C for 2 h with rotation at 600 rpm in a thermomixer compact (Eppendorf). 10 mM EDTA was then added to stop the enzymatic reaction. The cell suspension was passed through a 70 μm filter and stained with live/dead fixable violet stain (Thermofisher Scientific). Cells were subsequently stained with the following fluorescently conjugated antibodies; CD45 (30-F11), Ly6G (1A8), F4/80 (BM8), CD11b (M/170),

CD11c (N418), Thy1 (30-H12), Podoplanin (8.1.1.), PDGFRα (APA5; all from Biolegend) and CD31 (390; eBioscience) at 1:300 dilution. Flow cytometry was performed on LSR Fortessa (BD Biosciences) analyzers. Unstained and single-stained compensation beads (Invitrogen) were run alongside to serve as controls. Offline analysis was carried out on FlowJo (Treestar). For in vivo PAD4 inhibitor studies, tumors were separated for flow cytometric and immunofluorescent analysis. Some GSK484 treated tumors were too small for analysis and were therefore excluded (only tumor volumes were recorded).

**In vitro treatment of skin tumor cells with PAD4 inhibitors**. Tumor cells were isolated from skin tumors by digestion with 4 mg/ml Collagenase A for 1 h at 37 °C. The cells were then strained through a filter and seeded onto a culture dish in DMEM supplemented with 5% FBS and 1% penicillin/streptomycin. The cells were cultivated for 3–4d and then seeded onto 24-well plates at a density of $3 \times 10^4$ cells/well. The cells were either treated with DMSO (vehicle control), 100 μM Cl-amidine or 10 μM GSK484 for 48 h. Images were taken at 0, 24, and 48 h after treatment.

**In vitro neutrophil staining**. Bone marrow neutrophils were isolated as described above and treated with lung FB or CAF CMed for 30 min. Neutrophils were then treated with 10 μM 2′,7′-Dichlorodihydrofluorescein diacetate (DCFDA; Sigma-Aldrich) or $2 \times 10^8$ yellow/green 1 μm fluoresbrite beads (Polysciences Inc.) for 20 min. Alternatively, neutrophils were treated with CMed and stained with antibodies for CD11b (M/170), CD18 (M18/2) and CD62L (MEL-14; all from Biolegend). In some experiments, neutrophils were treated with CMed and stained with Annexin V (BD Pharmingen) and 7-AAD (Thermofisher Scientific) for 30 min. For all experiments, the cells were washed and then immediately analyzed by flow cytometry.

**PAD4 inhibitor treatment on neutrophil function in vitro**. Murine bone marrow neutrophils were seeded at $1 \times 10^5$ cells per condition in complete endothelial cell culture medium mixed 1:1 with pancreatic CAF CMed. Neutrophils were treated with or without 1.5 mg/ml Cl-amidine for 3 h. Neutrophil activation was then assessed by measuring expression of CD11b, CD18, and CD62L. ROS production was assessed by measuring DCFDA by flow cytometry (as above). The phagocytic activity of the neutrophils was assessed by measuring fluorescent bead uptake by flow cytometry. Neutrophil degranulation was determined by surface expression of CD35 and CD63.

**Characterization of CAFs and FB**. Isolated CAFs and FB were stained for typical markers; Podoplanin, PDGFRα and Thy1 (as described above) and markers to exclude immune cells (CD45), endothelial cells (CD31) and epithelial cells (EpCAM clone G8.8; all from Biolegend).

**Gene expression analysis**. Pancreatic CAFs were treated with ~10 ng/ml NETting DNA (from CAF CMed or PMA stimulated neutrophils) for 24 h. RNA was isolated using the RNeasy Mini Kit (Qiagen, Crawley, UK), converted to cDNA and analyzed by qPCR using Universal PCR mastermix (Life Technologies) according to manufacturer's instructions. Primers were bought as Assay on Demand kits from Applied Biosystems. qRT-PCR was performed using TaqMan assays (Col1a2 Mm00483888_m1, Acta2 Mm00725412_s1 and Gapdh Mm99999915_g1) and a StepOne Real-Time PCR System (both Life Technologies). Levels of each gene were expressed as $2^{-\Delta CT}$ (relative to Gapdh).

**Analysis of Amyloid-related genes**. Expression of app and bace2 and correlations with CAF and lymphatic markers in human pancreatic adenocarcinoma and skin cutaneous melanoma compared to healthy tissue were analyzed using GEPIA RNA-sequencing expression data taken from the publicly available TCGA and GTEx projects [http://gepia.cancer-pku.cn][75]. For mouse pancreatic tumor samples, publicly available data were analyzed (GEO accession number GSE42605).

**Immunofluorescence and tissue stains**. Tumors were snap frozen in OCT medium (TissueTek). 5–7 μm sections were fixed in ice-cold acetone/methanol for 5 min. For formalin-fixed paraffin-embedded murine lung tumors (provided by Daniel Munos-Espin, University of Cambridge), sections were first dewaxed in xylene and rehydrated in graded alcohols then subjected to antigen retrieval in Tris-EDTA pH9. Samples were blocked in 10% donkey serum in PBS and incubated with primary antibodies overnight at 4 °C. Primary antibodies as follows: Rabbit anti-citrullinated histone H3 R17 + R2 + R8 (ab5103, 1:500 dilution, Abcam), goat anti-myeloperoxidase (AF3667, 1:1000 dilution, RnD Systems), hamster anti-podoplanin (clone 8.1.1, 127402, 1:100 dilution, BioLegend), Alexa 488 conjugated mouse anti-APP (clone 22C11, MAB348A4, 1:100 dilution, Merck-Millipore), mouse anti-αSMA (clone 1A4, MAB1420, 1:100 dilution, RnD Systems). Slides were washed, incubated with appropriate secondary antibodies and counterstained with DAPI. Sections were mounted in Slowfade Gold (Invitrogen).

Images were acquired on a Leica SP5 confocal microscope. For visualization of collagen, frozen 10 μm sections of murine skin tumors were stained with Hematoxylin (Sigma) for 4 min then washed for 10 min under running tap water. Slides were stained with Herovici staining solution (prepared as described in ref. [76]) for 2 min. The sections were then rinsed in 1% acetic acid and dehydrated by placing slides in 100% ethanol until mounting. Slides were mounted in DPX (Sigma) and then imaged under a light microscope.

**Human samples**. The research was conducted in full accordance with the Guideline for Good Clinical Practice and the Declaration of Helsinki. All biological materials used in this study and their subsequent evaluations were in accordance with the informed consent agreements obtained from all subjects.

Human tissue and plasma samples from patients with advanced melanoma (AJCC clinical stage III and IV) were kindly provided by the MELRESIST study investigators (UK NRES committee North East – Newcastle & North Tyneside 2 Research Ethics committee review reference 11/NE/0312). Control plasma was isolated from consenting healthy volunteers from the Cambridge Blood Centre. Human tissue microarrays for pancreatic adenocarcinoma were obtained from a commercial supplier (Biomax.US) and additional local ethical approval was obtained for use of these TMAs in our study (HBREC 2019.16).

**Immunofluorescence of human tissue**. Formalin-fixed paraffin-embedded human pancreatic TMA slides (Biomax), and melanoma samples were dewaxed in xylene and rehydrated in graded alcohols prior to antigen retrieval in Tris-EDTA pH9. Slides were blocked and incubated with primary antibodies as above at 4 °C overnight. Samples were washed and incubated in fluorescently conjugated secondary antibodies before cell nuclei were counterstained with DAPI and sections were mounted in Slowfade Gold and imaged on a Leica SP5 confocal microscope.

**ELISA**. Levels of murine Amyloid β42, (Fisher Scientific), vWF and fibrinogen (both from Abcam) were measured in plasma collected from the skin and pancreatic tumor-bearing mice or in serum-free CMed from pancreatic CAFs as per manufacturer's instructions. Human platelet-poor plasma was obtained by centrifuging whole blood from healthy donors at $2000 \times g$ for 15 min. Samples were immediately snap frozen until use. Archived melanoma patient plasma samples (MELRESIST NRES: 11.NE.0312) and healthy controls were measured for levels of Amyloid β42 (Abcam) as per manufacturers guidelines.

**Mass spectrometry**. FB and CAF CMed were generated without fetal bovine serum for 24 h. The protein fractions were isolated and concentrated using 3 kDa centrifugal filters (Merck Millipore). LC MS/MS was performed on the concentrated culture CMed and spectral analysis was performed. Data were analyzed using Scaffold 4 software.

**Measurement of the effects of NETs or NET CMed on tumor cell and CAFs in vitro**. $1 \times 10^5$ neutrophils were treated with CAF CMed diluted 1:1 in complete EC media (Cell Biologics) with or without Cl-amidine and incubated for 3 h to generate NETs. The media was then harvested from the NETting or NET inhibited neutrophils. PBS supplemented with 1U/ml micrococcal nuclease (Sigma-Aldrich) was then added to the neutrophils for 10 min at 37 °C and 5% $CO_2$ to detach the NETs. The enzyme was inactivated with 0.5 mM EDTA. PBS containing the NETs was then harvested for xCELLigence assays.

Pancreatic CAF or tumor cells were seeded onto 16 well E-plates (ACEA Biologics) at a density of $5 \times 10^3$ cells/well. Cells were allowed to adhere for 1 h. The media was then replaced with the NETting or NET inhibited neutrophil CMed. Plates were placed into an xCELLigence RTCA MP Real-Time Cell Analyzer (ACEA Biologics). Recordings of the impedance, correlating to cell proliferation, were then taken over a 48 h period. The background was then subtracted from the appropriate wells.

**Statistical analysis**. Data are expressed as mean ± SEM, where a different neutrophil isolate and batch of CMed was used for each experiment. Multi-variant data were analyzed using analysis of variance (ANOVA), followed by Dunnett or Tukey post-hoc tests. Mann–Whitney or $t$-test was used to compare individual treatment conditions. $p < 0.05$ was considered statistically significant. For mass spectrometry, differences in protein expression in FB and CAF CMed was considered significant when the spectral count was $p < 0.01$. Data were analyzed using Graphpad Prism 7 and 8 Software packages.

**Reporting summary**. Further information on research design is available in the Nature Research Reporting Summary linked to this article.

## Data availability

Publicly available data were obtained from TCGA and the GTEx projects and the GEO repository (accession number GSE42605). The raw numbers for study data are available in the Source Data file wherever possible. Source data are provided with this paper.

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

## Acknowledgements

The authors would like to thank the staff at the ARES and CRUK Cambridge Institute animal facility for assistance with in vivo experiments, members of the CIMR flow cytometry core for assistance with flow cytometry applications, the Munos-Espin lab at the Hutchison/MRC Research Centre, the University of Cambridge for kindly providing the murine lung tumor sections, and Professor David Tuveson (Cold Spring Harbor Laboratories) for the murine pancreatic cancer cell line used in syngeneic models. This work was supported by the Medical Research Council Core funding. T.J. was supported by Cancer Research UK funding (C42738/A24868), Cold Spring Harbor Laboratory (CSHL) and Northwell Health, the Pershing Square Foundation, and the US National Institute of Health for funding received as part of Cancer Center Support Development Funds granted to CSHL (5P30CA045508-31). M.H. was supported by the German Research Foundation (DFG) (CRC1181-C03) and S.J.W. was supported by a MRC Clinical Academic Research partnership grant (G101982).

## Author contributions

J.D.S. and H.M. conceived the study, designed, performed, analysed and interpreted experiments, and wrote the manuscript. J.O.J. performed experiments and critically edited the manuscript. T.J. performed KPC experiments and provided advice and critically edited the manuscript. M.H. and M.E. provided PAD4KO and littermate control mice and provided advice and critically edited the manuscript. C.P.M. provided tissue from mice with lung tumors. S.J.W. provided valuable human tissue and plasma samples to support this work.

## Competing interests

C.P.M. is an employee of AstraZeneca. The other authors declare no competing interests.
