## [Peer Review File · Nature Communications]

REVIEWER COMMENTS

Reviewer #1 (Neutrophils/NET and cancer) (Remarks to the Author):

Dear. Dr. Shields,

It was a pleasure to read your manuscript entitled "Stromal driven Amyloid β -dependent induction of neutrophil extracellular traps modulates tumour growth". The manuscript is well written and convincing in its rationale and logical experimental progression. I believe this work is important and of high impact. Indeed, several teams including ours have found NETs to significantly impact cancer progression on many fronts and your work provides a novel and highly interesting molecular target by which these events are mediated in some important tumor sites. It is my opinion that the work merits publication in Nature Communications and I only have some minor comments that I would like to see addressed prior to publication.

1. While the data on lung CAFs and fibroblasts are interesting, they are not corroborated with any in vivo experiments addressing lung cancer models. Both murine and human data indicate that NETs play an important role in lung cancer progression. The absence of this data is unfortunate and one wonders if the team had a technical issue with assessing response to therapy in an orthotopic model or if there are other data that conflict with what was found for melanoma and pancreatic models.
2. With regards to the human data exploration in figure 6, data on lung cancer are conspicuously missing. Genetic information should be relatively easily sourced. Similarly, imaging of NETs in human lung cancers should be relatively feasible along with staining for Beta amyloid. If the team does not currently have access to such samples, they could be obtained via collaboration.
3. The summary figure could be improved. It does not effectively communicate how CAF-NET interactions support tumor growth. Even if this component is not the subject of this study, it should be represented along with our general understanding of these mechanisms. In essence, how this study fills an important gap in the context of our current knowledge around how NETs promote the growth of primary tumors would be more beneficial to understanding the value of the paper.
4. The authors should consider using the PAD4 KO mouse model to assess the hypothesis that the lack of effect of their GSK PAD4 inhibitor's failure is due to lack of tissue penetration. This could be accomplished via an orthotopic murine model of pancreatic in the PAD4KO versus WT mice. Resolving this issue of tissue penetration would be highly important to meaningfulness of this therapeutic avenue for pancreatic cancer patients.
5. Figure S4 is mislabelled.

Sincerely,

Jonathan Spicer

Reviewer #2 (CAF, EV, cancer) (Remarks to the Author):

Fig 1 - how confident are the authors that the PDPN positive cells represent CAF? Other markers? Why PDPN?

Why did PMA induce neutrophil death? Is that expected?

Any relationship between CAF and NET abundance in primary tumours? Data from human tumours?

Positive control for GCSF neutralisation (Fig 1G)?

Strange to suggest G-CSF plays a role in stimulation of NETs with CAF CMed infusion when Fig 1G suggests this isn't the mechanism (last para, p6)

Effect of GSK484 in vitro should be included in Fig 2 - does it block NET formation? Is there evidence for this in vivo? Fig 3 doesn't provide convincing evidence - strong conclusion 'Therefore, the effect of inhibiting PAD4 on the tumor growth in vivo was primarily due to inhibition of NETosis.' without a great deal of supporting evidence

Fig 4 - was APPalpha measured also, or just beta? Need to show BACEi actually reduces APPbeta secretion

Fig 5D - need quantitation to be able to state changes in expression at protein level - suggest amend text

Fig 6 - 'The high expression of bace2, the rate-limiting step in Amyloid β release, rather than app was correlated with poorer patient prognosis in both tumor types (Figure 6F and G)' - suggest amending wording here to make clear no correlation between APP levels and outcome

p13, last sentence: 'we reported a trend towards an increase in Amyloid β 14 levels in patients with melanoma compared to healthy volunteers. However, with our small cohort of archived patient samples, this was not significant'. I suggest removing reference towards a trend toward - the result was not significant, so this speculation should be avoided. It would be reasonable to say this is worth assessing in a larger cohort.

REVIEWER COMMENTS

We thank the reviewers for taking time to review our manuscript, for constructive suggestions that have led to a strengthened manuscript. We have addressed comments below.

Reviewer #1:

It was a pleasure to read your manuscript entitled “Stromal driven Amyloid β -dependent induction of neutrophil extracellular traps modulates tumour growth”. The manuscript is well written and convincing in its rationale and logical experimental progression. I believe this work is important and of high impact. Indeed, several teams including ours have found NETs to significantly impact cancer progression on many fronts and your work provides a novel and highly interesting molecular target by which these events are mediated in some important tumor sites. It is my opinion that the work merits publication in Nature Communications and I only have some minor comments that I would like to see addressed prior to publication.

We very much appreciate the kind words of reviewer 1.

1. While the data on lung CAFs and fibroblasts are interesting, they are not corroborated with any in vivo experiments addressing lung cancer models. Both murine and human data indicate that NETs play an important role in lung cancer progression. The absence of this data is unfortunate and one wonders if the team had a technical issue with assessing response to therapy in an orthotopic model or if there are other data that conflict with what was found for melanoma and pancreatic models.

We have added representative images showing neutrophils and NETs in the same lung cancer model from which the CAFs and bone marrow neutrophils were isolated in later experiments (Supplementary Figure 1F). As lung tumours tend to develop a significant CAF compartment only when they become very large, we do not believe the murine lung tumour model is the best for mechanistically studying the interactions of CAFs and neutrophils for disease progression. Thus, studies in subsequent figures were focussed on skin and pancreas. We have amended the text clarify this, which can be found on page 7.

2. With regards to the human data exploration in figure 6, data on lung cancer are conspicuously missing. Genetic information should be relatively easily sourced. Similarly, imaging of NETs in human lung cancers should be relatively feasible along with staining for Beta amyloid. If the team does not currently have access to such samples, they could be obtained via collaboration.

We agree with the reviewer regarding the human data. However, in light of the observations in point 1 we focussed on pancreatic and skin tumours for the remainder of the manuscript. Reflecting this observation, publicly available data on lung adenocarcinoma didn't show such strong correlations between APP and BACE2 expressions and have not been included. Furthermore, unfortunately as a consequence of COVID restrictions still in place on the hospital campus here, we have been unable to access lung tissue.

3. The summary figure could be improved. It does not effectively communicate how CAF-NET interactions support tumor growth. Even if this component is not the subject of this study, it should be represented along with our general understanding of these mechanisms. In essence, how this study fills an important gap in the context of our current knowledge around how NETs promote the growth of primary tumors would be more beneficial to understanding the value of the paper.

We thank the reviewer for this comment. The figure included is the graphical abstract of the data rather than a working hypothesis summary. We have amended the abstract to include the features we have shown for NET-treated CAFs, and have discussed how these changes may support tumour growth in the discussion section on Page 14.

4. The authors should consider using the PAD4 KO mouse model to assess the hypothesis that the lack of effect of their GSK PAD4 inhibitor's failure is due to lack of tissue penetration. This could be accomplished

via an orthotopic murine model of pancreatic in the PAD4KO versus WT mice. Resolving this issue of tissue penetration would be highly important to meaningfulness of this therapeutic avenue for pancreatic cancer patients.

Thank you for this suggestion. Based on the concern raised, we first utilized a subcutaneously implanted pancreatic tumour model as our animal licence does not enable us to perform surgical procedures needed for orthotopic implantation. Implanted cells were derived from KPC tumours presented in Supplementary Figure 5. Once C57Bl/6 mice developed established tumours, they were treated with the GSK484 as described for genetic models. Here, we observed a complete suppression of tumour growth (Supplementary Figure S6) akin to the response seen in the skin tumour model (Figure 3B). Additionally, we did not observe NETting neutrophils in the treated tumours and the bone marrow neutrophils were also less capable of undergoing NETosis. These data would suggest that the ability of the drug to penetrate the tumour was indeed the main stumbling block to the success of this assay originally presented.

To further validate these findings, we performed the same subcutaneous pancreatic tumour model in PAD4KO mice, as suggested by reviewer 1. Once again, we observed that the tumours did not grow in the absence of NETting neutrophils further emphasising the importance of this process in regulating tumour progression (Supplementary Figure S6). This data also confirms the findings in Supplementary Figure S4F where inhibition of PAD4 does not affect the growth of the tumour cells, because in these assays the implanted tumour cells were still capable of expressing PAD4.

5. Figure S4 is mislabelled.

Thank you for spotting this. This has been corrected.

.....
Reviewer #2:

Fig 1 - how confident are the authors that the PDPN positive cells represent CAF? Other markers? Why PDPN?

No single marker identifies all CAFs, or is 100% specific. However, a growing number of publications have shown that podoplanin is a highly robust CAF marker in pancreatic tumours and have used it to define CAFs for isolation and subset characterisation (including Eyada, E. et al. (2019) *Cancer Discovery*. 9(8): 1102-1123; Dominguez, C. et al. (2020) *Cancer Discovery* 10:232-53; and Hirayama, K. et al. (2018) *Surgery Today*. 48(1): 110-118). In light of this, we too used the same marker to capture as much of the CAF compartment as possible to assess whether neutrophils and NETs were in close proximity. With podoplanin, we reliably detect deposits of spindle shaped cells. From other work in the lab, these are not lymphatics and co-localise with other typical stromal markers.

Why did PMA induce neutrophil death? Is that expected?

It has previously been shown that PMA stimulation can induce neutrophil apoptosis (Saito, T. et al. (2005) *Biosci Biotechnol Biochem*. 69(11):2207-2212). In our hands we observed only 10% death after 30min stimulation which is expected, and in line with this report.

Any relationship between CAF and NET abundance in primary tumours? Data from human tumours?

This is an interesting question, but beyond showing their presence or absence in murine and human tumours, we are unable to measure NETs within tumours in a sufficiently quantitative fashion to be able to predict how t-NET abundance is related to primary tumour progression.

Other studies have shown that NET abundance in patient blood can influence cancer-associated thrombosis, a leading cause of death in cancer patients. In support of this we showed that pharmacological inhibition of NETosis reduced clotting, and presented smaller tumours.

Positive control for G-CSF neutralisation (Fig 1G)?

We used the same antibody used in a previous study that showed an important role for G-CSF driven NETosis in cancer-associated thrombosis (Demers, M. (2012). *PNAS*. 109(32): 13076–13081). We used the neutralising antibody at a concentration recommended by the manufacturer to completely neutralise G-CSF *in vitro*. Furthermore, G-CSF driven NETosis has mainly been implicated in systemic NETosis which is not the primary focus of the paper; we were more focussed in potential drivers of NET formation in solid tumours. Additionally, a recent study has shown that G-CSF induces vital NETosis (where the neutrophils survive after undergoing NETosis; Arpinati, L. et al. (2020) *Cancer Immunology Immunotherapy*. 69(2): 199-213), while we show that t-NETosis is a suicidal process. For these reasons we didn't pursue G-CSF as a candidate driver of t-NETosis any further.

Strange to suggest G-CSF plays a role in stimulation of NETs with CAF CMed infusion when Fig 1G suggests this isn't the mechanism (last para, p6)

We apologize that the description of this data was unclear. We are not suggesting that G-CSF stimulates NETosis in response to CAF-derived factors in any way. We speculate, in line with other reports (Singh, P. et al. (2012) *Leukaemia*. 26(11): 2375-83, Semerad C. L. et al. (2002) *Immunity*. 17(4): 413-23, Eyles, J. L. et al. (2008) *Blood*. 112 (13): 5193–5201, among many others) that it may support the increased number of neutrophils in bone marrow observed when we intravenously infused CAF CMed or in tumour bearing mice (data not shown). We have amended the text to clarify this (Page 6). Furthermore, it has previously been shown that human CAFs can secrete G-CSF in gastric cancers (Morris, K. T. et al. (2014) *British Journal of Cancer*. 110(5): 1211–1220).

It is widely accepted that G-CSF is a critical driver of neutrophil production. As such, G-CSF seems to be a logical candidate driver of the increased neutrophil number we observe in CAF CMed treated mice. Indeed, patients receiving chemotherapy also receive daily doses of G-CSF to boost neutrophil production and liberation from the bone marrow in an effort to prevent infection (reviewed by Bendall, L. J. et al. (2014) *Cytokine Growth Factor Review*. 25(4):355-67).

Effect of GSK484 *in vitro* should be included in Fig 2 - does it block NET formation? Is there evidence for this *in vivo*? Fig 3 doesn't provide convincing evidence - strong conclusion 'Therefore, the effect of inhibiting PAD4 on the tumor growth *in vivo* was primarily due to inhibition of NETosis.' without a great deal of supporting evidence

Thank you for this comment. As we concentrated on GSK484 inhibition studies *in vivo* and no *in vitro* assays were performed using GSK484, this data was not included. We have included new *in vivo* data performed in pancreatic tumours (Supplementary Figure S6E, and shown below). Here in the bone marrow, neutrophils produced NETs. In contrast, treatment with GSK484 abolished the capacity of bone marrow neutrophils to NET, indicating that their ability to NET was suppressed by PAD4 inhibition without affecting the total neutrophil number in the bone marrow. Furthermore, we didn't observe NETs in the subcutaneous pancreatic tumours treated with GSK484 (Supplementary Figure S6). Further experiments performed in PAD4KO mice, where host cells (but not implanted tumour cells) were deficient in PAD4, phenocopied the effects of pharmacological inhibition (Supplementary Figure S6F) confirming that blocking NETosis was the main cause for the reduction in tumour growth observed in these disease models.

Fig 4 - was APPalpha measured also, or just beta? Need to show BACEi actually reduces APPbeta secretion

We did not measure APP alpha in the CAF CMed as we showed that amyloid beta specifically, and not alpha, drove NETosis in our model (Supplementary figure 8C). We have also updated Figure 4F to show new data that amyloid beta production by CAFs is inhibited by BACEi treatment.

Fig 5D - need quantitation to be able to state changes in expression at protein level - suggest amend text

We thank the reviewer for this comment. In Figure 5E, we qualitatively show that there is more collagen deposition in the NET treated CAFs compared to untreated CAFs. However, we have amended the text to make this clearer on page 11.

Fig 6 - 'The high expression of bace2, the rate-limiting step in Amyloid β release, rather than app was correlated with poorer patient prognosis in both tumor types (Figure 6F and G)' - suggest amending wording here to make clear no correlation between APP levels and outcome

Thank you for spotting this. The text has been amended.

p13, last sentence: 'we reported a trend towards an increase in Amyloid β 14 levels in patients with melanoma compared to healthy volunteers. However, with our small cohort of archived patient samples, this was not significant'. I suggest removing reference towards a trend toward - the result was not significant, so this speculation should be avoided. It would be reasonable to say this is worth assessing in a larger cohort.

Thank you for spotting this. The text has been amended on page 14.

REVIEWERS' COMMENTS

Reviewer #1 (Remarks to the Author):

Thank you for your careful attention and responses to reviewer comments and questions. I have no further concerns regarding the publication of this manuscript.

Reviewer #2 (Remarks to the Author):

Thanks very much for addressing my comments. Although I'm not entirely convinced with some of the responses, a reasonable case has been made throughout, including the addition of supporting data, so there is no reason to further delay publication in my view. This is an interesting study that I'm sure generate plenty of interest. I'd like to congratulate the authors for this excellent work.